# *Gli1*-expressing stromal cells are highly reparative precursors of long-lived chondroprogenitors in the fetal murine limb

Xinli Qu[1,14], Ehsan Razmara[1,14], Ashiq Khader C[2], Chee Ho H'ng[1], Kailash K. Vinu[1], Luciano G. Martelotto [3,12], Maia Zethoven [4], Fernando J. Rossello[5,6,7], Shanika L. Amarasinghe [1,8,13], David R. Powell [8,9] & Alberto Rosello-Diez [1,2,10,11] ✉

The growth-plate cartilage of the developing long bones is a well-known system of spatially segregated stem/progenitor, transient amplifying and terminally differentiated cells. However, the regulation of the number and activity of long-lived cartilage progenitors (LLCPs) is poorly understood, despite its relevance for understanding human-height variation, the evolution of limb size and proportions and the aetiology of skeletal growth disorders. Moreover, whether their behaviour can adapt to developmental perturbations, generating robustness, has not been explored. Here, we show that Gli1+ cells are the fetal precursors of postnatal LLCPs, and that Gli1+ LLCP precursors remain mostly dormant until postnatal stages. However, in response to genetically-induced cell-cycle arrest targeted to the fetal cartilage, they expand in the cartilage, enabling normal growth. We further show that reparative Gli1+ cells originate from Pdgfra+ cells outside the cartilage, revealing the surrounding tissues as an unexpected CP source. Elucidating how stromal cells become Gli1+ LLCPs could shed light on developmental robustness and lead to growth-boosting therapies.

Developmental systems often display remarkable robustness—that is, the ability to overcome challenges that affect cell function or viability[1]. Elucidating the mechanisms underlying robustness could improve our fundamental understanding of how individual cells integrate information and coordinate with their neighbours, such that organ-level collective behaviours are achieved. Moreover, it could lead to growth-modulating therapies for congenital or acquired growth disorders.

Long bones in the limbs are a powerful model to study growth regulation. They grow via a transient cartilage template that is progressively replaced by bone[2]. Cartilage cells (chondrocytes) undergo progressive end-to-centre differentiation, starting as resting/reserve chondrocytes that are progressively recruited into the proliferative pool (flat cells arrayed in columns), which eventually enlarge and differentiate to hypertrophic chondrocytes. Some hypertrophic

[1]Australian Regenerative Medicine Institute, Monash University, Melbourne, VIC, Australia. [2]Department of Physiology, Development and Neuroscience, University of Cambridge, Cambridge, UK. [3]University of Melbourne Centre for Cancer Research, The University of Melbourne, Melbourne, VIC, Australia. [4]Peter MacCallum Cancer Centre, Melbourne, VIC, Australia. [5]Murdoch Children's Research Institute, The Royal Children's Hospital, Melbourne, VIC, Australia. [6]Novo Nordisk Foundation Center for Stem Cell Medicine, Murdoch Children's Research Institute, Melbourne, VIC, Australia. [7]Department of Clinical Pathology, University of Melbourne, Melbourne, VIC, Australia. [8]Monash Genomics and Bioinformatics Platform, Monash University, Melbourne, VIC, Australia. [9]Monash eResearch Centre, Monash University, Melbourne, VIC, Australia. [10]Department of Genetics, University of Cambridge, Cambridge, UK. [11]Loke Centre for Trophoblast Research, University of Cambridge, Cambridge, UK. [12]Present address: Adelaide Centre for Epigenetics, South Australian immunoGENomics Cancer Institute, University of Adelaide, Adelaide, SA, Australia. [13]Present address: Biomedicine Discovery Institute, Monash University, Melbourne, VIC, Australia. [14]These authors contributed equally: Xinli Qu, Ehsan Razmara. ✉e-mail: ar2204@cam.ac.uk

chondrocytes die, while others transdifferentiate into osteoblasts (bone-laying cells) that replace the cartilage with bone, forming the primary ossification centre[3,4]. Osteoblasts can also migrate, along with blood vessels, from the perichondrium[5], a connective tissue layer that surrounds the cartilage and has been shown to contain a stem-cell niche known as the groove of Ranvier[6]. A secondary ossification centre (SOC) forms later (~postnatal day 7 in the mouse) at both ends of the skeletal element, so that the growth cartilage gets 'sandwiched' into a so-called growth plate by both ossification centres.

Maintaining the size and structure of the growth plate—in other words, the balance between cartilage generation and destruction—is critical to sustain bone growth over time and requires refined control mechanisms. The size of the proliferative zone is maintained approximately constant due to a well-described negative feedback loop between Indian hedgehog and Parathyroid hormone-related peptide that balances proliferation and differentiation[3]. But other control mechanisms are less known, especially regarding the resting-to-proliferative transition. In fetuses and neonates, cartilage progenitors (CPs) located in the reserve zone do not self-renew[7], so that when one is recruited into a proliferative column, its lineage gets rapidly exhausted, leading to short clonal columns. At approximately 3 weeks of age, CPs start to self-renew—possibly in response to SOC signals—giving rise to long clonal columns (Supplementary Fig. 1a)[7]. These long-lived CPs (LLCPs) are the key driver of subsequent growth and therefore of final bone length. Therefore, elucidating their origin and regulation is critical to understanding normal and compensatory growth, as well as skeletal growth disorders and the evolution of limb size. One of the biggest unresolved questions is whether postnatal LLCPs arise from fetal short-lived ones, in response to intrinsic and/or extrinsic triggers (Supplementary Fig. 1a), or whether they are already present but inactive in the fetal limb, becoming activated (or even recruited from outside the cartilage) in response to cartilage maturation (Supplementary Fig. 1b). Determining which of these mechanisms takes place during normal growth, and whether they can adapt in response to developmental perturbations, is desirable for the development of targeted growth therapies, focussed on stimulating the expansion of the pool of LLCPs at the right time and location. In this study, we probe the system using models of cartilage-targeted cell-cycle arrest, combined with lineage tracing and functional analyses. We find that Gli1+ cells in the fetal limb (including outside the cartilage) give rise to dormant CPs, which later postnatally give rise to most chondrocytes of the growth plate. Moreover, these progenitors can be precociously activated by cartilage-targeted growth challenges, revealing significant plasticity and a strong reparative potential. We posit that this fundamental knowledge will be critical to understanding and manipulating cartilage growth and repair.

## Results

### Cell-cycle arrest in the fetal cartilage triggers a compensatory response from Gli1+ cells

We previously generated a fetal mouse model of left cartilage-targeted, mosaic (i.e. salt-and-pepper) overexpression of p21 (which blocks the cell cycle in G1 phase[8]), by combining *Pitx2-Cre*[9], *Col2a1-rtTA*[10] and *Tigre^Dragon-p21* alleles[11] (Fig. 1a, *Left-Cart-p21^MOE* model hereafter). In that study, we showed that the defect was compensated by hyperproliferation of the spared (p21−) chondrocytes[11]. We reasoned that, if short-lived CPs were the only source of LLCPs (as in Supplementary Fig. 1a), their hastened proliferation would lead to premature exhaustion of the CP pool (Supplementary Fig. 1c), and therefore to reduced bone length in the long term. To test this hypothesis, we followed mice from the *Left-Cart-p21^MOE* model until postnatal day (P) 100, well beyond the end of the growth period. Interestingly, no major asymmetries in bone length were found as compared to *Pitx2-Cre/+; Tigre^Dragon-p21/+* mice (Controls, Ctl hereafter, Supplementary Fig. 1d, the left/right ratio was not significantly different for the tibia, and only

~1.5% lower for the femur). This result suggested that either the spared resident CPs in the *Left-Cart-p21^MOE* acquired self-renewing properties precociously, or that facultative CPs (i.e. cells that would not normally contribute to cartilage at that age) were recruited to the challenged cartilage. To shed light on this process, we undertook an unbiased single-nuclei RNA-seq approach of left and right fetal knees from two Ctl and two *Left-Cart-p21^MOE* embryos (Fig. 1a and Supplementary Fig. 2a). From the original dataset (~32,000 nuclei in 32 clusters, Supplementary Fig. 2b, d), which contained mostly lateral-plate-mesoderm-derived cells, and a minority of muscle and endothelial cells, we further explored the former population (~21,000 nuclei, Supplementary Fig. 2d'). This subset was composed of 23 clusters (Fig. 2c, d'), which we grouped in five main categories (see Methods): chondrocytes (Supplementary Fig. 2b'), tenocytes, fibroblasts, perichondrium and joint-associated mesenchymal cells. Analysis of regulon activity with SCENIC[12] (Supplementary Fig. 3a, Supplementary Data 1), complemented by our previous bulk RNA-seq analysis of the *Left-Cart-p21^MOE* model[11] (Supplementary Fig. 3b, b') revealed that the activity of GLI-Kruppel family member 1 (GLI1) transcription factor was significantly upregulated in the left experimental (ExpL) osteochondroprogenitors and to a lesser extent in resting chondrocytes (Fig. 1a'). This was interesting because Gli1+ osteochondroprogenitors have been shown to participate in the repair of bone fractures[13]. Thus, we next asked whether the number of Gli1+ cells changed in response to cartilage-targeted p21 overexpression. Indeed, bioinformatics analysis[14] revealed that, within the Gli1^high lateral-plate-mesoderm-derived population, the proportion of cells belonging to the chondrocyte group was increased in experimental limbs, as compared to Ctl (Supplementary Fig. 2e, e'). To confirm these findings, we turned to Cre-based lineage tracing. To trace the Gli1-lineage independently of p21 expression, we utilised a Doxycycline (Dox)-controlled, Cre-independent p21 expression system, targeted to all cartilage elements (*Col2a1-rtTA/+; Tigre^TRE-p21/+, Pan-Cart-p21^MOE* model hereafter; where TRE means Tet-responsive element)[15]. Females bearing a tamoxifen (TM)-inducible Cre recombinase (CreER^T2) cassette[16] targeted to the *Gli1* locus (*Gli1^CreER* hereafter) were crossed with *Pan-Cart-p21^MOE* males bearing also a Cre-reporter cassette (lox-STOP-lox-tdTomato) targeted to the safe harbour locus *Rosa26* (*R26^LSL-tdTom* hereafter)[17] and given Dox at E12.5 and TM at E13.5 (Fig. 1b, b'). As with the model targeted to the left limb, we confirmed that expression of p21 in 60-70% of E14.5 chondrocytes (Supplementary Fig. 4a) did not cause growth defects by postnatal day (P) 0 (Fig. 1b'). To assess the involvement of Gli1+ cells in this compensatory response, we followed the distribution of tdTom+ cells inside and outside the cartilage at 1, 4 and 7 days post TM (Fig. 1c, c'). In *Pan-Cart-p21^MOE* mice, the Gli1-lineage showed a significant cartilage expansion, as compared to controls, at all stages analysed (Fig. 1c'). These results suggested that, in control conditions, some chondrocytes derived from fetal Gli1+ cells are only transient and eventually eliminated from the cartilage, while in *Pan-Cart-p21^MOE* limbs, the Gli1-lineage increases its contribution to longer-lived CPs. Given that CPs reside in the resting zone[18], we divided the cartilage analysis into resting and proliferative, finding that both zones showed an increased number of tdTom+ cells, typically biased towards the resting zone (Supplementary Fig. 4b-b''). Moreover, we observed that the population of cycling chondrocytes (i.e. Ki67+) in *Pan-Cart-p21^MOE* mice got enriched in Gli1-lineage cells, as compared to control mice (Supplementary Fig. 4c). Of note, the contribution outside the cartilage was also expanded in P0 *Pan-Cart-p21^MOE* limbs compared to controls (Fig. 1c'), suggesting that the challenged cartilage signals to the surrounding tissues.

Importantly, since our initial labelling of the Gli1-lineage was done after the induction of p21 expression (Fig. 1b'), we were not able to distinguish between two scenarios: (i) all the Gli1+ cells derive from formerly Gli1+ cells; (ii) *Gli1* expression is activated de novo in cells that were Gli1− before. Since the second scenario would reveal an unknown

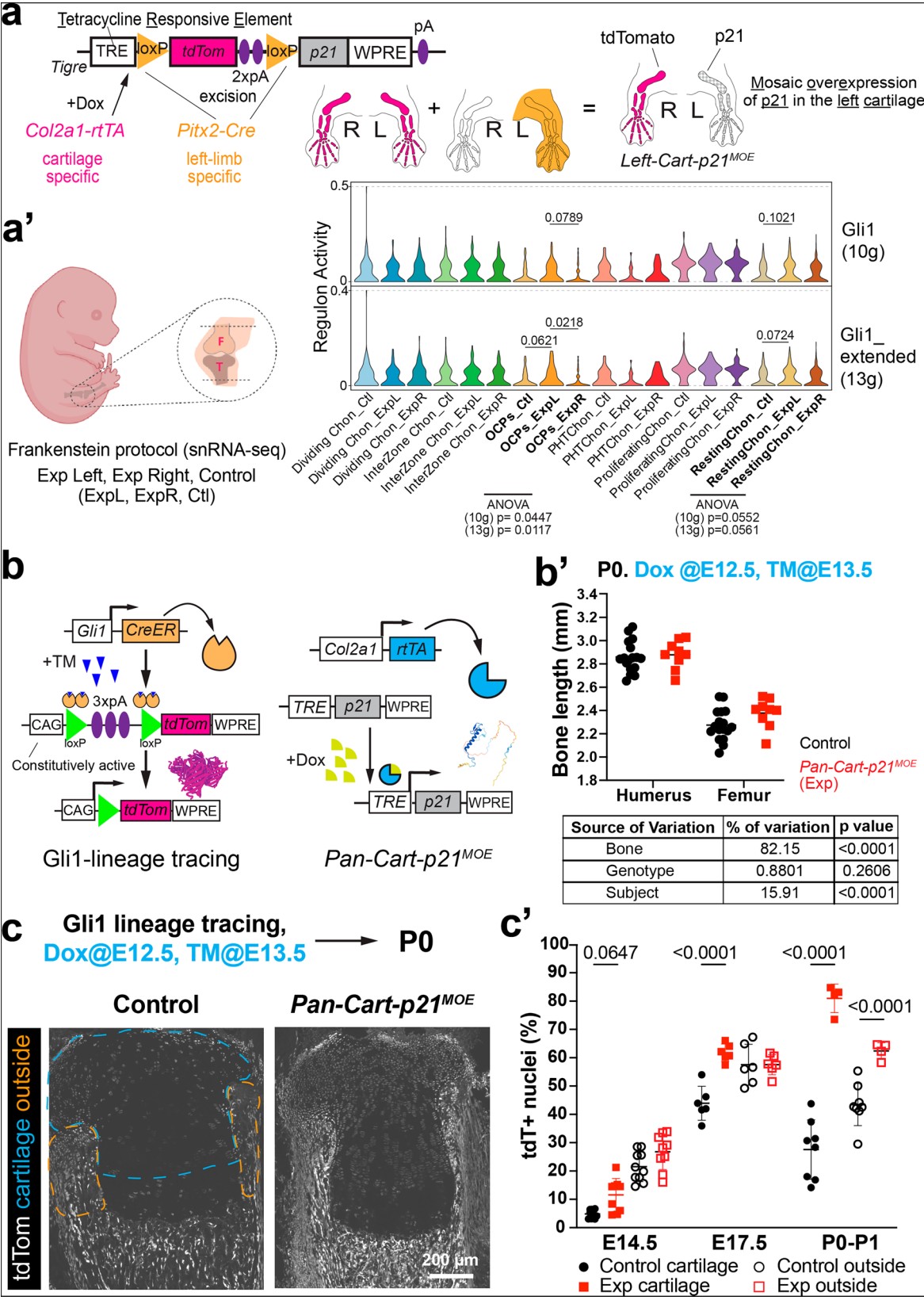

degree of plasticity that could be relevant for the understanding of growth control and the design of future therapies, we decided to further investigate this matter. We reasoned that, by providing TM-first, and Dox 1 day later, most of the tdTom-labelling would happen among existing Gli1+ cells, with TM clearance mostly taking place before any potential de novo *Gli1* expression. Indeed, it was previously shown with

a cartilage-driven CreER that recombination is complete 24 h after TM injection[19]. Using this TM-first approach, we found that the Gli1-lineage expansion was not as dramatic as in the Dox-first approach, becoming significant only by P0 (Supplementary Fig. 4d). Moreover, within P0 *Pan-Cart-p21^MOE* animals, the number of Gli1 descendants in the cartilage was lower in TM-first as compared to the Dox-first approach,

**Fig. 1 | Mosaic cell-cycle arrest in the fetal cartilage triggers compensation from Gli1+ stromal cells. a, a′** A left-specific, cartilage-targeted p21 mosaic over-expression model (*Left-Cart-p21^MOE*) was used for snRNA-seq analysis of knees. TRE, Tet-responsive element. WPRE, RNA-stabilising sequence. Left (L) and right (R) knees were processed separately but analysed together in control embryos (**a**). Chondrocyte-related cells were analysed via SCENIC. A focus on Gli1 activity is presented as violin plots (**a′** right). *p* values of ANOVA (across conditions within each cell type) are tabulated in the Supplementary Table 1. Those ≤0.1 are also shown on the graph. Tukey-corrected *p* values for multiple comparisons tests are also shown if ≤0.1. Expression of p21 in the cartilage, induced from E12.5 to P0 (**b**), does not lead to growth defects (**b′**, *n* = 17 Ctl, 9 Exp). The table shows the results of 2-way ANOVA. **c, c′** The Gli1-lineage was followed in the absence (Control) and presence of p21 misexpression (*Pan-Cart-p21^MOE*), and quantified in the regions shown in (**c**). In (**c′**), data shown as mean ± SD. *p* values for Sidak's multiple comparisons test after 2-way ANOVA are shown (*n* = 11, 6, 11 Ctl; 9, 6, 4 Exp).

especially in the proliferative zone (Supplementary Fig. 4e). This result suggests that while the existing population of Gli1+ cells contributes to the repair, some Gli1− cells become Gli1+ underline{after} *p21* induction in the cartilage (Supplementary Fig. 4f). To further test this possibility, we quantified the proportion of Gli1+ cells within tdTom-negative cells in the TM-first condition. We found that the number of cells that were initially Gli1-negative but subsequently gained expression of *Gli1* increased in the experimental condition as compared to control littermates (Supplementary Fig. 4g). This effect was not significant at E14.5 (2 days after Dox addition), but it was at E17.5 (5 days after Dox addition), suggesting that the response takes some time to build up. Along these lines, the proportion of Gli1+ cells that belonged to the Gli1-lineage did not change between control and experimental samples when the lineage was labelled before p21 expression was triggered, but it did increase in experimental samples when p21 expression was triggered before the Gli1-lineage was labelled (Supplementary Fig. 4h).

We next characterised the expression of *Gli1* in its own traced lineage, in both control and *Pan-Cart-p21* samples. We found that *Gli1* mRNA expression is only detected in less than 40% of the E13.5-traced chondrocytes (tdTom+), when analysed at E14.5 and E17.5 (Supplementary Fig. 5a′), suggesting that, in normal growth, *Gli1* expression is quite transient in many chondrocytes. However, in *Pan-Cart-p21^MOE* littermates, significantly more lineage-traced chondrocytes retained *Gli1* expression at both stages (Supplementary Fig. 5a′), suggesting that this could be relevant for the observed compensation.

To test whether Gli1-derived chondrocytes are required for the compensation of *Pan-Cart-p21^MOE*, we set to specifically kill them using a *Tigre^Dragon-aDTA* allele in which inducible expression of attenuated diphtheria toxin (aDTA) requires Cre and rtTA activity[15,20] (Fig. 2a). Mice were crossed as shown in Fig. 2a to generate *Gli1^CreER/+; Col2a1-rtTA/+; Tigre^TRE-p21/+* embryos (phenotypically similar to *Pan-Cart-p21^MOE*), *Gli1^CreER/+; Col2a1-rtTA/+; Tigre^Dragon-aDTA/+* (*Gli-Cart-DTA* model) and *Gli1^CreER/+; Col2a1-rtTA/+; Tigre^Dragon-aDTA/ TRE-p21* littermates (*Pan-Cart-p21^MOE; Gli-Cart-DTA* model, Fig. 2a), in addition to control (rtTA−) ones. As shown in Fig. 2a, a′, we gave Dox to induce p21 expression in the cartilage (*Pan-Cart-p21^MOE*), and TM to induce cell death of the rtTA-expressing Gli1-derived chondrocytes (*Gli-Cart-DTA* combination). As predicted, TUNEL staining revealed apoptosis only in the cartilage of *Gli-Cart-DTA* and *Pan-Cart-p21^MOE; Gli-Cart-DTA* embryos (Fig. 2b, b′). We then measured bone length at P0 and found that only *Pan-Cart-p21^MOE; Gli-Cart-DTA* pups showed significantly decreased bone length as compared to *Pan-Cart-p21^MOE* and/or control pups (Fig. 2c). This result demonstrated that Gli1-derived chondrocytes are required to compensate for cell-cycle arrest in the fetal cartilage, although it did not rule out more subtle effects in *Gli-Cart-DTA* pups, which showed a trend towards decreased bone length. Later time points could not be analysed due to the high mortality of animals carrying the *Gli1-Cart-DTA* genotype.

### Foetal Gli1+ cells give rise to most chondrocytes in the juvenile and adult growth plate

Having identified a highly-reparative fetal Gli1+ population, we set out to determine their role during normal growth. One study showed that when Gli1+ cells are lineage-traced at 1 month of age, they label 60-80% of chondrocytes of the growth plate 1 month later[21], indicating that P30 Gli1+ cells include long-lived chondroprogenitors (LLCPs). We thus

tested whether fetal Gli1+ cells (present inside and outside the cartilage; Supplementary Fig. 5a) already contained LLCPs. We traced Gli1-lineage cells from either embryonic day (E) 12.5 or E13.5, and quantified the tdTom+ cells at E14.5, E17.5, E19.5, P30 and P60. In all cases, tdTom+ cells were found both inside the cartilage and adjacent tissues, including perichondrium, metaphysis, interzone knee region and SOC (Fig. 3a, a′). Since fetal CPs had been shown not to self-renew[7], we expected the % of lineage-traced chondrocytes to decrease with time. On the contrary, the contribution of Gli1-lineage cells to the cartilage increased with time, labelling up to 65–70% of chondrocytes in the 2-month-old growth plate (Fig. 3b). To ascertain when Gli1-derived chondrocytes became LLCPs (i.e. switched from forming short to forming long clonal columns), we used a tri-colour reporter line (RGBow, Fig. 3c)[22] that randomly expresses one of three membrane-bound fluorescent proteins upon Cre-recombination. After giving TM at E12.5 or E13.5, we analysed the number, location and length of distinct columns in the three regions of the cartilage. Notably, the total number of columns increased up to P0-P1, the majority located in the resting and proliferative zones (Fig. 3c, c″) and composed of 2–3 cells (Fig. 3d). By P14, the total number of columns was significantly reduced, but started to become longer (Fig. 3c″, d). By P30, few columns remained, but many crossed the whole growth plate, starting from the RZ (Fig. 3c′, c″). In summary, these results suggested that Gli1+ cells at E12.5 or E13.5 give rise to CPs with two different fates: short-lived clones that get depleted over time, and long-lived ones that remain mostly dormant at perinatal stages and become more active after the second postnatal week. To further probe the existence of these two populations, we performed label-retaining experiments with the thymidine analogue 5-ethynyl-2′-deoxyuridine (EdU). We injected EdU at E13.5 and E14.5 (60 mg/kg each day, to reach saturation) to females carrying *Gli1^CreER/+; R26^LSL-tdTom/+* embryos (given TM at E12.5), and provided 5-chloro-2′-deoxyuridine CldU at the times of collection (P3, P7, P14) to quantify proliferative activity at those times (Supplementary Fig. 5b). As expected, the Gli1-lineage in the RZ became enriched in EdU-retaining cells between P7 and P14 and, in fact, most EdU-retaining cells belonged to the Gli1-lineage by P14 (Supplementary Fig. 5b′, b″). Moreover, EdU-retaining cells showed a peak of proliferation at P7 (Supplementary Fig. 5b″), supporting the notion that an early wave of CPs proliferates and abandons the RZ early postnatally, and what remains is the LLCP population, which is mostly quiescent at perinatal stages.

### Chondrocytes generated in the last prenatal week from Gli1+ cells are required for long-term bone growth

To test if Gli1-derived chondrocytes are required to sustain normal bone growth, we crossed *Gli1^CreER; Tigre^Dragon-aDTA* mice with *Col2a1-rtTA; R26^LSL-tdTom*, and gave TM at E13.5 and Dox E12.5-P0. This way, only chondrocytes were ablated among Gli1 descendants (Fig. 4a–b′). While the ablation was incomplete and only sustained for a short time, this targeted injury led to reduced bone length by P100, especially for the femur and tibia (Fig. 4c). This revealed the important role of fetal/perinatal Gli1-derived chondrocytes in long-term bone growth.

### Long-lived CPs are derived from Gli1+ cells that continuously seed the fetal and perinatal cartilage

We next asked whether Gli1-derived LLCPs were already present in the early cartilage template (around E12.5), or whether they were

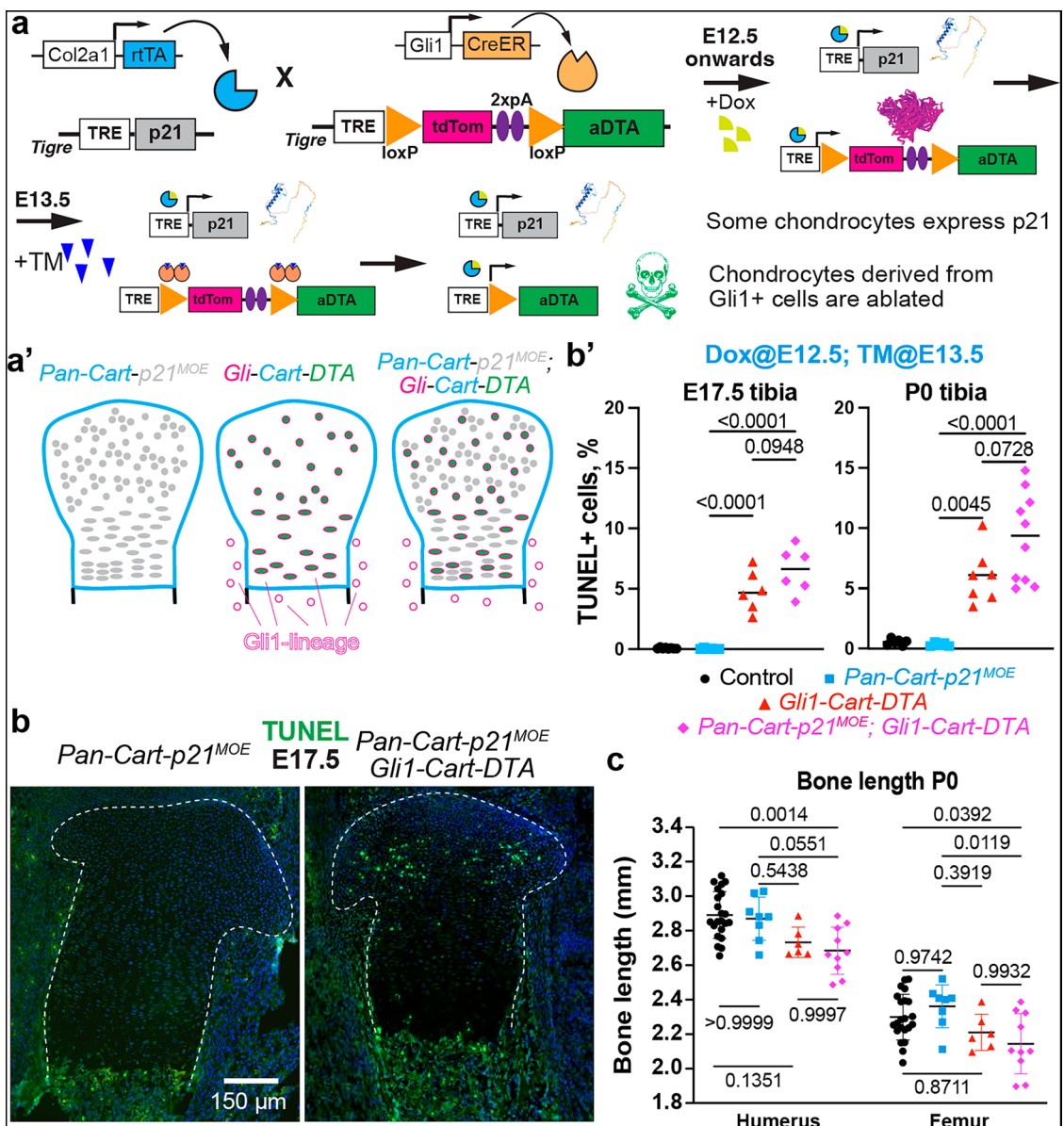

**Fig. 2 | Gli1-derived chondrocytes are needed for compensation of *Pan-Cart-p21^MOE*.** A *Gli1^CreER* allele and *Col2a1-rtTA* are combined with *Dragon-aDTA* (**a**, aDTA= attenuated diphtheria toxin) and/or *TRE-p21* to specifically ablate Gli1-derived chondrocytes in normal growth and p21-induced compensation (**a'**). Example images of TUNEL (**b**, cartilage outlined) and quantification (**b'**) for the indicated stages and genotypes (*n* = 8, 6, 6 at E17.5, *n* = 6, 5, 7, 10 at P0). Significant *p* values for Tukey's test after ANOVA are shown. **c** Length of the P0 mineralised region is shown for the indicated genotypes as mean ± SD (*n* = 17, 9, 6, 10). Significant *p* values of Sidak's multiple comparisons test (after 2-way ANOVA) are shown.

incorporated later from the adjacent tissues. The latter possibility is supported by studies showing that the chicken perichondrium is a reservoir for cartilage precursors[23]. To this end, we generated a mouse line in which expression of nuclear-directed GreenLantern fluorescent protein depends on Dox-activated rtTA (Fig. 5a and Supplementary Fig. 6a, b, *Tigre^TRE-nGL* hereafter). We then crossed *Gli1^CreER/CreER; Tigre^TRE-nGL/TRE-nGL* females with *Col2a1-rtTA/+; R26^LSL-tdTom/LSL-tdTom* males, and provided TM at E13.5 to label the Gli1-lineage with tdTom, and Dox for a 1-day pulse to transiently express nGL primarily in fetal chondrocytes. We then followed the green signal over time, with the expectation that quiescent chondrocytes (cells expected to become LLCPs) would retain fluorescence for longer than cells dividing quickly (Fig. 5a). Moreover, by providing Dox at different points, we aimed to label different potential CP populations (Fig. 5a'): those already resident in the early cartilage anlage (Dox at E12.5), and those seeded later into the cartilage (Dox at E17.5). Most green cells after E12.5-Dox were found

inside the cartilage, with a negligible number in the perichondrium (Fig. 5a'', Supplementary Fig. 6c, d), confirming that the new tool could be used to track cells located only in the cartilage. As predicted, the number of nGL⁺ chondrocytes dropped fast at first (Supplementary Fig. 6e), but then stabilised at 2–3% of the RZ cells at P14 and P21 (Fig. 5b, no green signal in the PZ), becoming undetectable by P30. Remarkably, the % of Gli1-lineage chondrocytes in the nGL⁺ population increased over time in the RZ (Fig. 5d), until ~90% of GL-retaining CPs at P14 belonged to the E13.5-labelled Gli1-lineage. Conversely, only ~3% of the Gli1-lineage cells in the P14 RZ were also nGL⁺ (Fig. 5e), confirming that the Gli1-lineage contains label-retaining and non-retaining subpopulations. Of note, the scarcity of nGL⁺ cells within the P14 Gli1-lineage could theoretically be due to the existence of early Gli1-derived CPs that are nGL⁻ despite being located in the cartilage and that expand later. To test this possibility, we co-stained sections of E14.5 limbs with an HCR probe for *Matn1*, as an independent cartilage

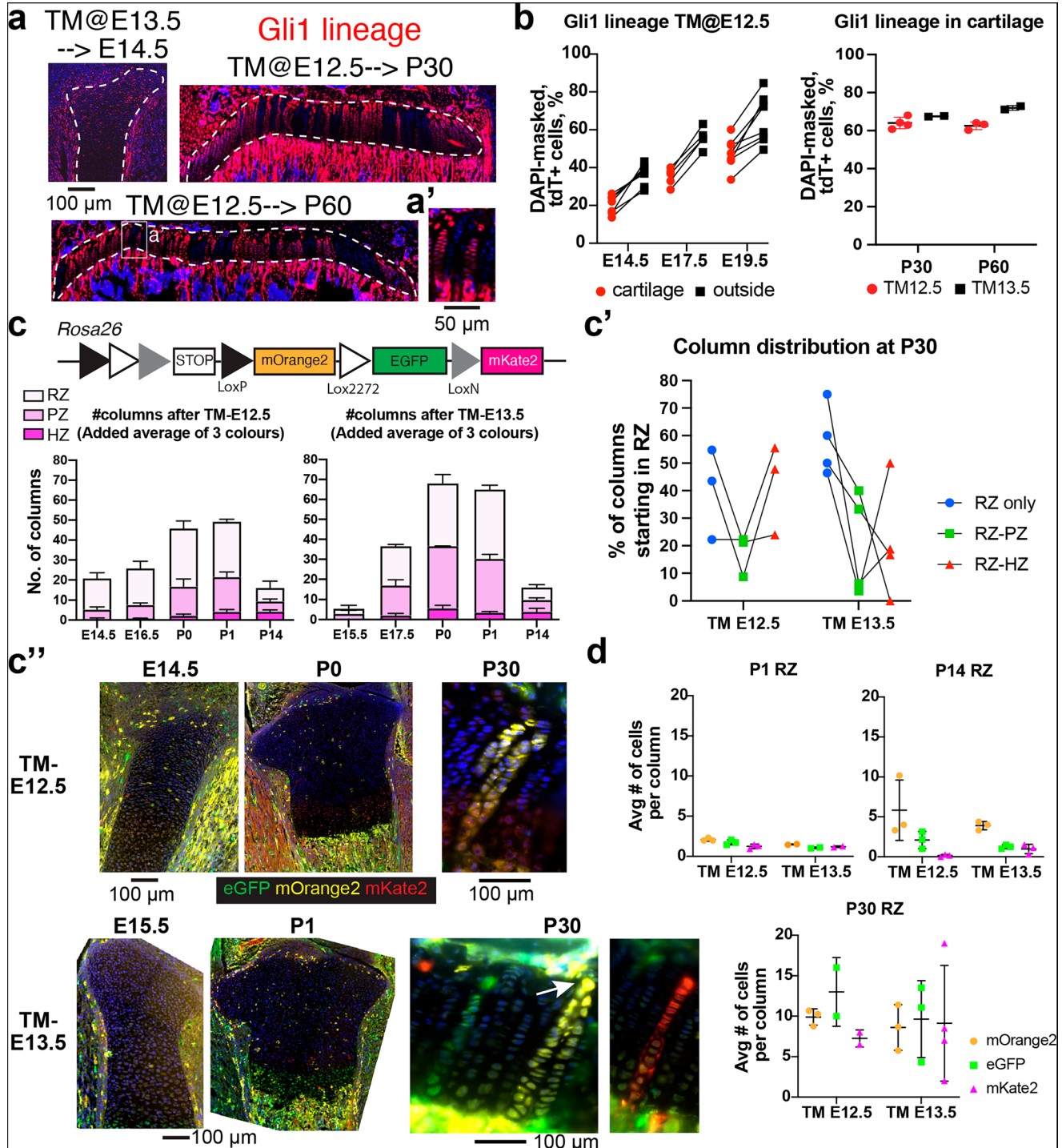

**Fig. 3 | Foetal Gli1⁺ cells give rise to most chondrocytes in the juvenile and adult growth plate. a** Examples of Gli1-lineage tracing after TM induction at E12.5 or E13.5, analysed at the indicated stages. **b** Quantification of reporter-expressing cells, inside and outside the cartilage for TM-E12.5 (left, $n = 6, 5, 6$), or in cartilage for both inductions (right, $n = 4, 2$, average and SD are shown). **c–c"** The tri-colour reporter RGBow was used to quantify individual clones in the growth plate (**c**, added average of each colour is shown), at multiple times post induction ($n = 3, 4, 3, 3, 3, 3$ for TM-E12.5, $n = 3, 3, 2, 2, 2, 3, 3$ for TM-E13.5). RZ, PZ, HZ: resting, proliferative, hypertrophic zone. In (**c'**), the % of P30 columns starting from RZ are shown, based on where they end ($n = 3$ and 4). **c"** shows representative images. **d** Number of cells per RZ-rooted column, for the stages and induction times indicated ($n = 3$ in all groups). Bars in (**c, d**) represent SD.

marker[24]. As shown in Supplementary Fig. 6d, we could not detect almost any tdTom⁺ nGL⁻ cells within the *Matn1*⁺ region, ruling out that possibility. When Dox was given between E17.5 and E18.5 (Fig. 5b', c, d', e'), the result was similar to the E12.5-Dox pulse, except for the proportion of nGL⁺ cells within the Gli1-lineage RZ chondrocytes, which was higher (~6.7% at P21) than when Dox was given at E12.5 (~2%). In

other words, while a 1-day Dox pulse may not be enough to fully activate nGL expression in all possible rtTA⁺ cells, the same pulse duration, given at E17.5, led to a different quantitative outcome at P21 as compared to E12.5. These results suggested that while many LLCPs are already present in the E12.5 cartilage, more LLCPs are added later, in both cases derived mostly from fetal Gli1⁺ cells. To test this

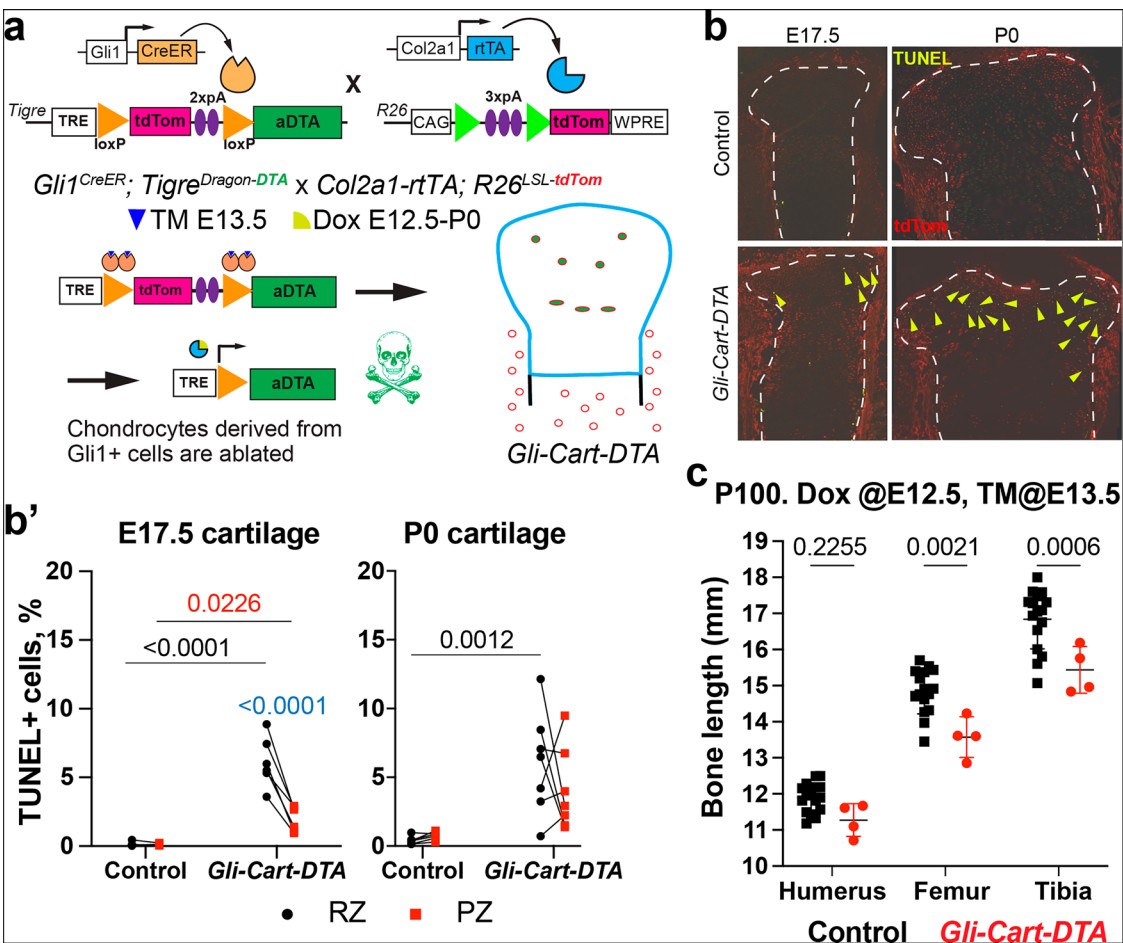

**Fig. 4 | Foetal-Gli1-derived chondrocytes are required for normal growth. a** The *Dragon-aDTA* allele was combined with *Gli1^CreER^* and *Col2a1-rtTA* to activate diphtheria toxin expression exclusively in Gli1-lineage chondrocytes in the last week of gestation (*Gli-Cart-DTA* model). Typical images (**b**) and quantification (**b'**) of TUNEL⁺ cells (arrowheads) at the indicated stages (*n* = 3, 6 at E17.5; 6, 7 at P0). **c** Bone length of Ctl and *Gli-Cart-DTA* mice at postnatal day 100 (P100, *n* = 17 Ctl, 4 Exp). Mean and SD are shown. In (**b'**, **c**), *p* values of multiple comparisons tests after 2-way ANOVA are shown. RZ, PZ: resting, proliferative zone.

prediction, we extended the Dox pulse from E12.5 to E18.5, so that all potential chondrogenesis waves were labelled in that period. Indeed, many more double-positive cells were found in the P14 RZ than with any other regimen (Fig. 5b'', c, d'', e''), in line with the theory that Gli1⁺ stromal cells initially located outside the cartilage get incorporated into the cartilage during the last week of gestation.

## Most Gli1+ LLCPs generated upon challenge derive from stromal Pdgfra+ cells

Having found evidence suggesting that stromal (non-cartilage) Gli1⁺ cells can become LLCPs (Fig. 5) and that some Gli1⁻ cells become Gli1⁺ upon p21 misexpression (Supplementary Fig. 4), we next set out to determine the origin and regulation of Gli1⁺ cells in normal and challenged growth.

To capture the potential arising of new Gli1⁺ chondrocytes from stromal cells, we chose *platelet derived growth factor alpha* (*Pdgfra*) as a marker of stromal cells that is virtually not expressed in the cartilage beyond E13.5 (Supplementary Fig. 5c, d, Supplementary Fig. 7). We crossed *Pdgfra^CreER/+^* females[25] with *Pan-Cart-p21^MOE^*; *R26^LSL-tdT/LSL-tdT^* males and provided Dox at E12.5 and TM at E13.5. We found that, in normal growth, the E13.5-labelled Pdgfra-lineage gave rise to an increasing proportion of chondrocytes (Fig. 6a, 30–40% at E14.5, 45–50% at E17.5 and ~60% at P0), strongly suggesting that the surrounding tissues can contribute to cartilage. Moreover, the proportion of Pdgfra-lineage chondrocytes was progressively increased in *Pan-Cart-p21^MOE^* limbs as compared to controls (Fig. 6a, compare P0 with E14.5). Of note, the

Pdgfra-lineage located in the resting zone showed increased proliferation in *Pan-Cart-p21^MOE^* vs. control limbs at E14.5 and E17.5 (Supplementary Fig. 8a–b'), potentially explaining the progressive expansion observed at and after E17.5 (Fig. 6a).

To determine to what extent Pdgfra-lineage expansion was related to the expression of *Gli1* in cartilage cells, we analysed the overlap of tdTom (marking the Pdgfra-lineage) and *Gli1* expression in cartilage cells at E17.5, under both TM-first and Dox-first induction regimens (Fig. 6b–d'). In the TM-first condition (early labelling of Pdgfra⁺ cells), *Pan-Cart-p21^MOE^* exhibited an increased proportion of double-positive cells (*Gli1*⁺ within the Pdgfra-lineage; Fig. 6c, triangles) at the expense of Gli1⁻ Pdgfra-lineage cells (Fig. 6c, circles), suggesting a shift within the lineage towards a *Gli1*⁺ identity. In contrast, the proportion of Gli1⁺ cells outside the Pdgfra-lineage remained largely unchanged (Fig. 6c, squares). Under the Dox-first regimen, where cell-cycle arrest is initiated earlier, Gli1⁺ cells within the Pdgfra-lineage were similarly increased in *Pan-Cart-p21^MOE^* samples (Fig. 6c', triangles), but without a corresponding reduction in the Gli1⁻ fraction (Fig. 6c', circles). Instead, there was a significant increase in Gli1⁺ cells not originating from the Pdgfra-lineage (Fig. 6c', squares), indicating either the very fast gain of *Gli1* and loss of *Pdgfra* expression in Pdgfra⁺ cells before they were exposed to TM, or the recruitment of Gli1⁺ cells from alternative sources. These results support a model in which a subset of Pdgfra⁺ cells in the surrounding tissue can lose *Pdgfra* expression and instead acquire *Gli1* expression under normal conditions, transitioning into Gli1⁺ CPs. This latent plasticity appears to be enhanced following p21-

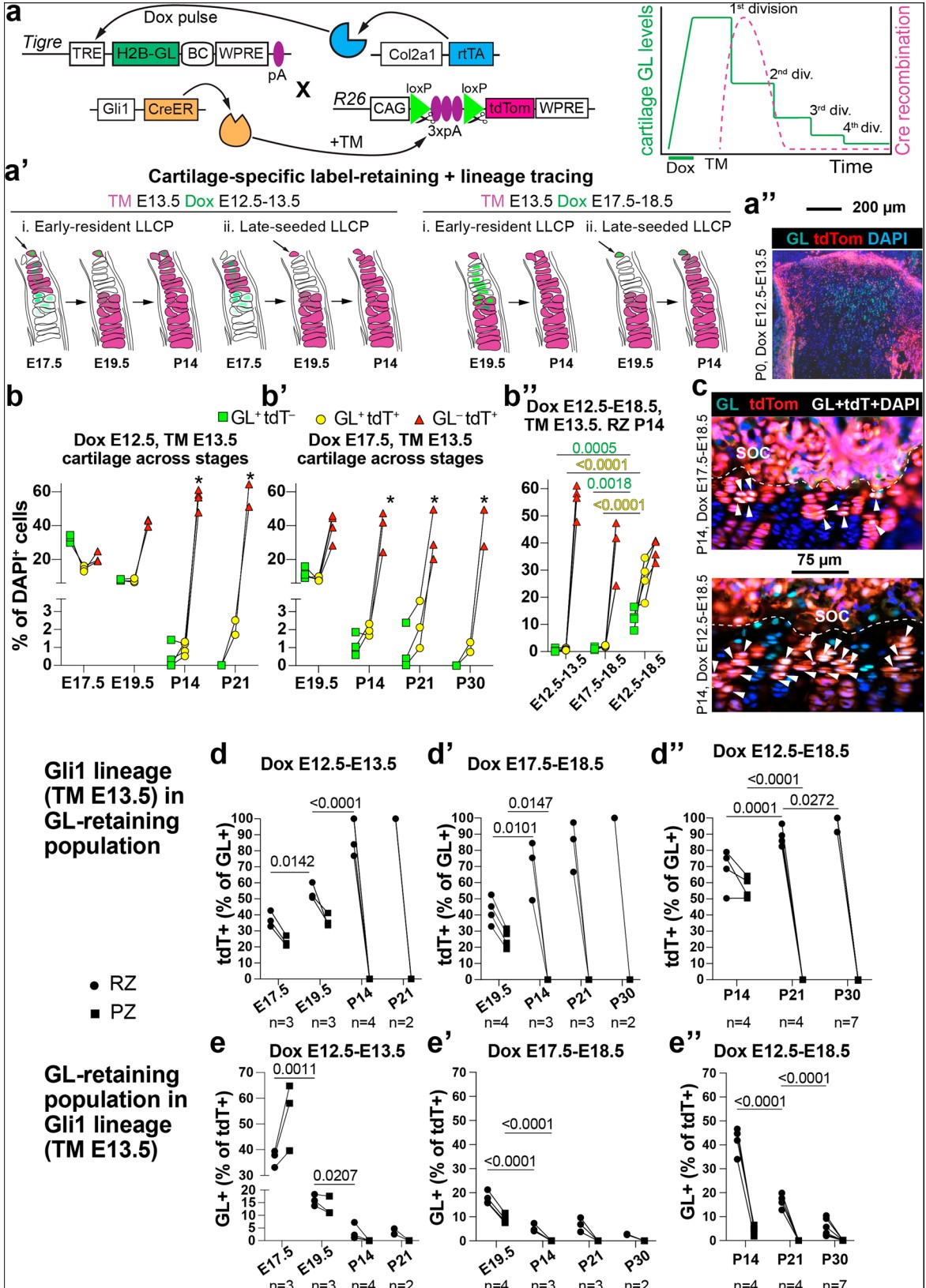

induced cell-cycle arrest in the cartilage, particularly when *Pdgfra* expression has already declined before lineage labelling (i.e. in the Dox-first regimen), leading to a greater contribution from Gli1+ cells outside the original lineage (Fig. 6d).

Although *Pdgfra* mRNA levels are low and transient in the cartilage (Supplementary Fig. 5c, d, Supplementary Fig. 7), we reasoned that

chondrocytes that had recently formed from Pdgfra+ cells in response to cell-cycle arrest would retain PDGFRA protein for some time. Therefore, to further explore Gli1+ chondrocytes as derivatives of the Pdgfra-lineage, we analysed PDGFRA protein expression within and outside the Gli1-lineage at E14.5, i.e. shortly after p21 induction (Fig. 6e). Indeed, the number of PDGFRA+ cells was increased within

**Fig. 5 | Most fetal LLCPs belong to the Gli1-lineage.** Model to permanently label the Gli1-lineage and transiently label chondrocytes with a nuclear GreenLantern (H2B-GL, nGL) that gets halved every cell division (**a**); different outcomes depending on Dox timing (**a′**) and a representative image of the distribution after early labelling (**a″**). BC, barcoding region (not used here). Quantification of single-positive and double-positive populations in the cartilage at the indicated stages after induction at E12.5-E13.5 (**b**, $n = 3, 3, 4, 2$), E17.5–E18.5 (**b′**, $n = 4, 3, 3, 2$) or E12.5–E18.5 (**b″**, $n = 4$). *: quantified in resting zone only. **c**, Representative images of

the Gli1-lineage cartilage in *Pan-Cart-p21^{MOE}* as compared to controls (Fig. 6e′, circles). In fact, the population that expanded the most in the *Pan-Cart-p21^{MOE}* vs. control cartilage was the PDGFRA⁺ Gli1-lineage one (Fig. 6e′), suggesting that this population is especially involved in the repair. To test this, we crossed *Pdgfra^{CreER/+}; Tigre^{Dragon-aDTA/+}* females with *Cart-p21* males. We provided TM at E12.5 and Dox at E13.5, so that Pdgfra-derived chondrocytes were ablated in the context of *Cart-p21* background (Supplementary Fig. 8c). As predicted, while the number of *Gli1⁺* cells increased in *Cart-p21* embryos, the number dropped to control levels when Pdgfra-derived chondrocytes were ablated (Supplementary Fig. 8c). In summary, our data suggests that Pdgfra⁺ cells (mostly located outside the cartilage) become Gli1⁺ CPs and progressively increase their contribution to cartilage in *Pan-Cart-p21^{MOE}* limbs as compared to controls, contributing to the compensation. Lastly, we examined the behaviour of the Pdgfra-lineage in long-term normal growth. Tracing of fetal Pdgfra⁺ cells with the RGBow reporter during normal growth led to very few long columns being labelled at P30 (Fig. 6f, compare with 3c′), with most of the few long columns being restricted to the periphery of the growth plate (Supplementary Fig. 8d–e′). These results suggest that Pdgfra⁺ cells do not give rise to a significant number of Gli1⁺ LLCPs unless the cartilage is challenged (Fig. 6g), leading us to question what molecular changes were triggered by the challenge.

### CCN2 impairs Gli1 expression and proliferation in chondrocytes, and is downregulated in Cart-p21 samples

To determine how the *Pan-Cart-p21^{MOE}* challenge triggers *Gli1* upregulation, we performed another transcriptomic approach, but this time shortly after challenge induction, separating Gli1-lineage and non-Gli1-lineage cells (Supplementary Fig. 9a). Pregnant females carrying *Gli1^{CreER/+}; R26^{LSL-tdTom/+}; Tigre^{TRE-p21/+}* embryos (with or without *Col2a1-rtTA*) were provided with Dox at E12.5 and TM at E13.5, and embryos were collected at E14.5. The whole hindlimb was processed for single-cell isolation and 10x Chromium library preparation, multiplexed by sex ($n = 4$ Exp and 4 Ctl samples, limbs from 2–3 embryos were pooled per sample, see Methods). Coarse clustering was initially generated and further refined manually (Supplementary Fig. 9b, b′). Supporting the connection between perichondrium and cartilage lineages, in the coarse clustering, the two resting chondrocyte subpopulations formed a continuum with the cells showing markers of the perichondrial groove of Ranvier, a niche that hosts cells that can make a contribution to cartilage[6]. To identify gene signatures changed in *Pan-Cart-p21^{MOE}* limbs, we performed pseudobulk differential gene expression analysis —which minimises false discovery rates[26]—via edgeR (Fig. 7a). Besides the expected cell-cycle related genes (e.g. histone-encoding), the second-biggest change was observed in a gene related to chondrocyte activity and response to stress, namely *cellular communication network factor 2* (*Ccn2*), also known as *connective tissue growth factor*[27–29], which was downregulated in *Pan-Cart-p21^{MOE}* limbs (see Violin plot and Gene Ontology analysis in Supplementary Fig. 9c, d). We complemented these results by analysing cell-cell communication via MultiNicheNet[30] in the E14.5 scRNA-seq dataset. In both RZ and PZ, *Ccn2*-mediated communication was significantly downregulated in experimental samples (Supplementary Fig. 9e, e′). We confirmed this using *Ccn2* hybridisation chain reaction (HCR) on *Pan-Cart-p21^{MOE}* and control samples (Fig. 7b, b′). Of note, this reduction happened in both Gli1⁺ and

tdT and nGL overlap (arrowheads) at P14 with 2 different induction regimes. SOC, secondary ossification centre (separated from cartilage by dashed lines). % of tdT⁺ cells within the nGL⁺ population (**d**–**d″**) or nGL⁺ cells within the tdT⁺ population (**e**–**e″**) after a 1-day Dox pulse at E12.5 (**d**, **e**) or E17.5 (**d′**, **e′**) or after a 6-day pulse at E12.5 (**d″**, **e″**). In all graphs $p$ values for multiple comparisons tests after 2-way ANOVA are shown. P21 data from (**d**, **d′**) and P30 data from (**e**, **e′**) were omitted from the analysis due to low sample size.

Gli1⁻ chondrocytes, suggesting that it was not a cell-autonomous consequence of *Gli1* activation (Fig. 7c, c′). These results were interesting considering our previous study in which we showed that overexpression of *Ccn2* in the left cartilage (*Pitx2-Cre/+; Col2a1-tTA/+; Tigre^{Dragon-Ccn2/+}* model, Supplementary Fig. 10a) impairs bone growth[15]. Using archival samples of this model, we observed that the left hypertrophic zone was extremely reduced in size (Supplementary Fig. 10a′), and that cycling cells (Ki67⁺) in the resting zone was significantly downregulated in the left cartilage as compared to the right one or control littermates (Supplementary Fig. 10b, b′). Interestingly, by using the ImageJ tool Distance Analysis (DiAna)[31], we also observed that the remaining Ccn2⁺ cells tended to be closer to p21⁻ cells than to p21⁺ ones (Supplementary Fig. 10c), suggesting that p21⁺ chondrocytes generate a Ccn2-inhibiting area.

Given the results above, and prior evidence that CCN2 orchestrates various signalling pathways to promote harmonised skeletal growth[32], we hypothesised that CCN2 interferes with *Gli1* activation in CPs, so that downregulation of *Ccn2* facilitated the expansion of Gli1⁺ cells in the *Pan-Cart-p21^{MOE}* models. To test this possibility, we cultured fetal femurs of control and *Pan-Cart-p21^{MOE}* fetuses ex vivo (see Online methods) and treated them with human CCN2 (one femur from each embryo) or Vehicle (the other femur from each embryo) for ~2 days (Fig. 7d). As predicted, *Gli1* expression was found downregulated in treated samples—both Ctl and *Pan-Cart-p21^{MOE}*—(Fig. 7d′). Moreover, Ki67 immunostaining was used to detect cycling cells, which were found to be diminished in treated samples (Fig. 7e, e′). In summary, we concluded that the expansion of the Gli1-lineage and the compensatory proliferation triggered by cartilage cell-cycle arrest were in part due to decreased CCN2 levels (Fig. 7f).

## Discussion

Elucidating the origin and regulation of the LLCPs is crucial to understanding the evolution of limb size and proportions, the factors that influence human height and the potential causes and treatments of skeletal dysplasias. In mice, postnatal CPs have been described that can self-renew and give rise to long columns of chondrocytes for most or the whole growth period[21,33,34]. However, given that at fetal and perinatal stages only short columns are generated[7], it was unclear whether LLCPs are already present (albeit inactive) in the early cartilage, or they are formed later. In either case, we hypothesised that the system likely evolved with built-in robustness, such that transient developmental challenges may reveal regulatory details of the process. Following this approach, we found that fetal Gli1-derived chondrocytes expand in response to cell-cycle arrest in the cartilage and are required to compensate for this challenge (Figs. 1 and 2). We further found that Gli1⁺ cells give rise to LLCPs during normal growth (Fig. 3) and that chondrocytes arising from the Gli1-lineage in the last gestational week are critical for acquiring normal bone length in the hindlimb (Fig. 4). While bone length was significantly affected when Gli1-derived chondrocytes were ablated, the effect was somewhat mild. This was probably due to compensation after Dox clearance. We tried to test this possibility by providing Dox postnatally, but the rtTA that we used in our models did not get sufficiently activated. Of note, while we showed that the early cartilage anlage already contains Gli1⁺ quiescent CPs, our results also suggested that Gli1⁺ cells outside the initial cartilage anlagen contribute to the LLCP pool later, at least until birth (Fig. 5).

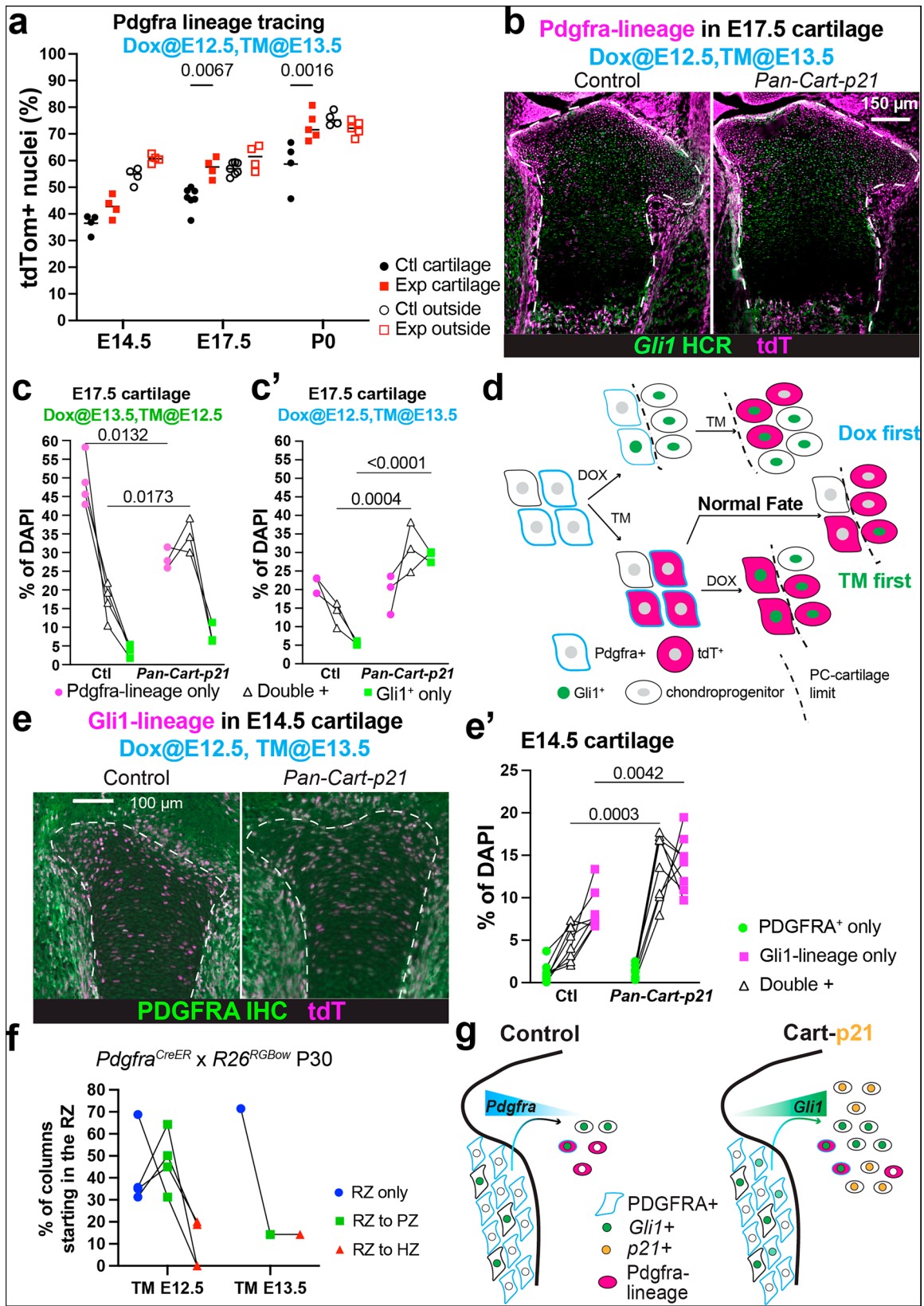

**a** Pdgfra lineage tracing
Dox@E12.5,TM@E13.5

**b** **Pdgfra-lineage** in E17.5 cartilage
Dox@E12.5,TM@E13.5

**c** E17.5 cartilage
Dox@E13.5,TM@E12.5

**c'** E17.5 cartilage
Dox@E12.5,TM@E13.5

**d** Dox first / Normal Fate / TM first

**e** **Gli1-lineage** in E14.5 cartilage
Dox@E12.5, TM@E13.5

**e'** E14.5 cartilage

**f** *Pdgfra^{CreER}* x *R26^{RGBow}* P30

**g** Control / Cart-p21

Interestingly, our single-cell RNA-seq analyses identified a continuous cluster including two resting chondrocyte populations and one of cells expressing markers of the groove of Ranvier (Supplementary Fig. 9b), with one of the resting populations within the Gli1^{high} subset being expanded at the expense of the groove-of-Ranvier one in *Pan-Cart-p21^{MOE}* samples (Supplementary Fig. 11a, b).

It is important to note, however, that this approach is limited because the Gli1-lineage had to be separated by FACS instead of bioinformatically identified, because in the 10x approach, which is 3′ biased, both the tdTom and the p21 transgenes yield reads mapping mostly to WPRE, the RNA-stabilising sequence used in both alleles (Fig. 1b).

**Fig. 6 | The contribution of the Pdgfra-lineage to Gli1⁺ LLCPs increases upon cartilage challenge. a** The Pdgfra-lineage was traced in the absence and presence of *Pan-Cart-p21^MOE^*, both inside and outside the cartilage (dashed lines; *n* = 4 Ctl, 4 Exp at E14.5; 6, 4 at E17.5; 4, 5 at P0). **b** Representative E17.5 image of Pdgfra-lineage⁺ and/or Gli1⁺ cells (*n* = 3 Ctl, 3 Exp). Quantification of the Pdgfra-lineage⁺ and/or *Gli1*⁺ populations in the E17.5 cartilage, under TM-first (**c**, *n* = 4 Ctl, 3 Exp) or Dox-first strategies (**c′**, *n* = 3 Ctl, 3 Exp). **d** Schematic model illustrating the contribution of Pdgfra-lineage and non-lineage cells to Gli1⁺ chondroprogenitors under different induction regimens. Representative image (**e**) and quantification (**e′**) of Gli1-lineage⁺ and/or PDGFRA⁺ cells in the E14.5 cartilage (*n* = 5 Ctl, 6 Exp). **f** RGBow-expression was induced in Pdgfra⁺ cells at either E12.5 (*n* = 4) or E13.5 (*n* = 1) and followed till P30. RZ, PZ, HZ: resting, proliferative, hypertrophic zone. The % of columns starting from RZ are shown, based on where they end. In (**a, c, c′, e′**), *p* values for Sidak's posthoc test after 2-way ANOVA. **g** Potential model of how Pdgfra-lineage moving into the cartilage could become Gli1⁺ LLCPs in response to *Pan-Cart-p21^MOE^*.

Regarding the origin of the Gli1⁺ CPs, our results suggested that some chondrocytes derive from stromal Pdgfra⁺ cells (Fig. 6), revealing a previously unknown source of chondrocytes. Moreover, we showed that these Pdgfra⁺ cells do not give rise to LLCPs during normal growth, but that they can become *Gli1* CPs (and maintain PDGFRA expression at least briefly) in response to cartilage challenge. This 'facultative role' would fit with the fact that *Pdgfra* loss in the developing limb does not have a major phenotype[35]. Future studies will address whether there is a chemoattractant signal emanating from the cartilage, capable of recruiting Pdgfra⁺ cells. Such a signal could have therapeutic potential if it could be exogenously provided to stimulate CP formation in situ, in a minimally invasive manner. In summary, our data support a model in which external cells contribute to the cartilage, but mostly form short-lived CPs, unless they switch their behaviour in response to a growth challenge. It is worth noting that previous studies used *Ctsk-Cre* to label most cells of the perichondrial groove of Ranvier, finding little[36] to no[37] contribution to the growth plate from P14 or P7 onwards, respectively. This aligns well with our finding that, in the absence of challenge, the Pdgfra-lineage almost did not contribute to the central growth plate from P14 onwards (Supplementary Fig. 8c).

Lastly, regarding the signalling changes operating in the challenged limbs, one of the first candidates we considered was the hedgehog signalling pathway, as *Gli1* is a hedgehog readout. In fact, the repair of large-scale rib fractures is another experimental paradigm in which cells surrounding the skeletal element (namely, a subpopulation of periosteal cells) are responsible for repair, and they do so in response to increased levels of sonic hedgehog[38]. In our case, however, differential gene expression and cell-cell communication analyses did not point to the hedgehog pathway, but showed that CCN2 levels were decreased in response to cartilage cell-cycle arrest (Fig. 7, Supplementary Fig. 9c). Subsequently, we showed that excess CCN2 impairs *Gli1* expression, chondrocyte proliferation and growth plate cytoarchitecture (Fig. 7 and Supplementary Fig. 10), leading us to propose a model in which CCN2 acts as a negative feedback signal (Fig. 7f). In this model, CCN2 levels reflect the growth status of the cartilage, such that reduced growth output leads to decreased CCN2 and thus increased number of Gli1⁺ CPs, therefore unleashing long-term bone growth potential. Future studies will focus on elucidating how growth status is linked to CCN2 levels, and how CCN2 and additional factors could be used to convert stromal cells into Gli1⁺ LLCPs, which could provide new avenues to stimulate cartilage regeneration in situ.

## Methods
### Animal models
The use of animals in this study was approved by the Monash Animal Research Platform animal ethics committee at Monash University. All mouse strains were outbred to a SwissWebster background for multiple generations before being used in the presented experiments. Females were bred from 7 weeks of age onwards and limited to 8 litters. Males were bred from 8 weeks of age onwards and to a maximum of 70 weeks of age. Mice were kept in individually ventilated cages (IVC), in rooms with a 12 h light cycle (6 am to 6 pm), at 22-24 °C and 40−60% humidity.

**Unilateral p21 misexpression model.** The *Pitx2-ASE-Cre* (aka *Pitx2-Cre*) mouse line, obtained from Prof. Hamada[9], was crossed with the *Col2a1-rtTA* mouse line, obtained from Karen Posey[10] and then inter-crossed to generate *Pitx2-Cre/Cre; Col2a1-rtTA* mice. In some experiments, a *Col2a1-eCFP* allele[39] was also included (a kind gift from David Rowe). Genotyping was performed as described previously[11]. The *Tigre^Dragon-p21^* mouse line was previously described[11,15]. Experimental and control animals were generated by crossing *Pitx2-Cre/Cre; Col2a1-rtTA/+* females with *Tigre^Dragon-p21/Dragon-p21^* males (i.e. homozygous for the conditional misexpression allele). The separation of control and experimental animals was based on the rtTA genotype. Pregnant females were administered doxycycline hyclate (Sigma) in water (1 mg/kg with 0.5% sucrose for palatability) from E12.5 until collection (or until birth). The day of vaginal plug detection was designated as E0.5, and E19.5 was referred to as P0.

**Multi-colour lineage tracing.** *Gli1^CreER/CreER^* female mice[40] were obtained from Bryce Vissel and time-mated with *R26R^RGBow/RGBow^* male mice (obtained from Tobias Merson). The day of vaginal plug detection was referred to as E0.5, and E19.5 as P0. Pregnant females were administered Tamoxifen (Sigma, stock at 20 mg/ml in corn oil) by oral gavage (180 µg/g) at E12.5 (Experiment 1) or E13.5 (Experiment 2). Mouse embryos were collected at E14.5, E16.5, P0, P1, P14, P30 (Experiment 1) or E14.5, E15.5, E17.5, P0, P1, P14, P30 (Experiment 2). Limbs were dissected and fixed in 4% paraformaldehyde for 2 d at 4 °C. Decalcification was performed on P14-P30 samples by immersing them in 0.45 M ethylenediaminetetraacetic acid (EDTA) in PBS from 10−14 days at 4 °C.

**Pulse of nuclear Green Lantern expression in cartilage.** *Gli1^CreER/CreER^; Tigre^TRE-nGL/TRE-nGL^* female mice were time-mated with *Col2a1-rtTA/+; R26^LSL-tdTom/LSL-tdTom^* male mice. Experimental set 1: Pregnant females were administered Dox in drinking water and Dox food (Specialty feeds SF08-026, 600 mg/Kg) from E12.5-E13.5. Tamoxifen was administered by oral gavage (180 µg/g) at E13.5. Mouse embryos were collected at E14.5, E17.5, P0, P14, P21. Experimental set 2: like set 1, except that Dox was given from E17.5−E18.5, and offspring was collected at P0, P14, P21 and P30. Experimental set 3: Like set 2, except that Dox was given from E12.5−E18.5, and offspring collected at P14. Limbs processed as above.

**Lineage tracing combined with p21 expression in all cartilage elements.** *Gli1^CreER/CreER^* females were crossed with *Col2a1-rtTA/+; Tigre^TRE-p21/TRE-p21^; R26^LSL-tdT/LSL-tdT^* males (described in refs. [15,17]). Pregnant females were given Dox food and water, plus tamoxifen (180 µg/g orally), at the stages described in the main text. To trace other lineages, *Pdgfra^CreER^* females[25] (a kind gift from Brigid Hogan) were used instead of *Gli1^CreER^* ones (tamoxifen 100 µg/g). *Pdgfra^H2B-EGFP^* mice[41] were a gift from Philippe Soriano.

**Ablation of Gli1-derived chondrocytes.** *Gli1^CreER/CreER^; Tigre^Dragon-aDTA/Dragon-aDTA^* females were crossed with males bearing one copy of the *Col2a1-rtTA* allele, with (Fig. 2) or without (Fig. 4) two copies of the *Tigre^TRE-p21^* allele. Of note, ablation could not be sustained postnatally in Fig. 4 experiment due to insufficient activity of *Col2a1-rtTA*, raising the possibility that partial compensation had taken place postnatally.

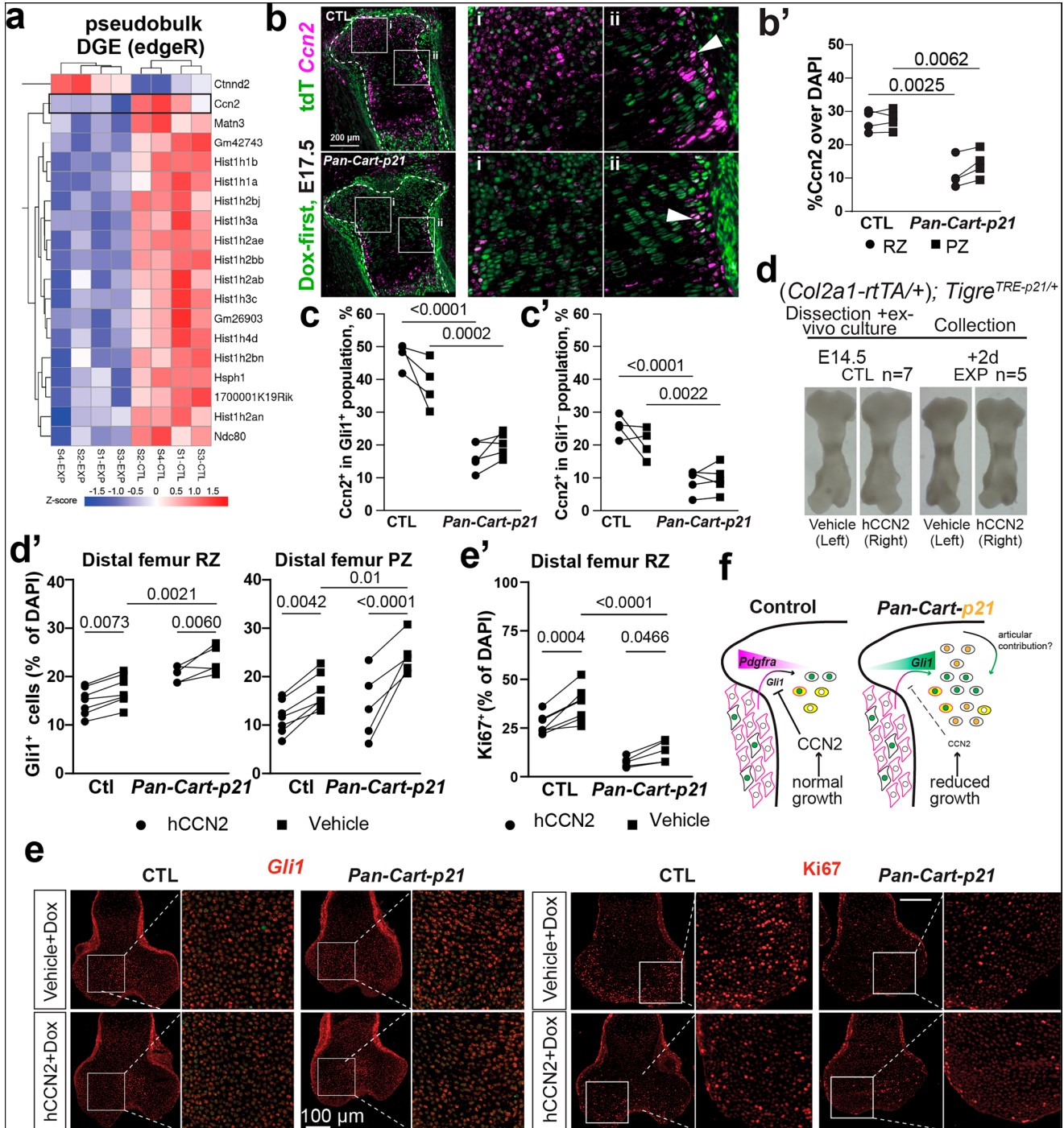

**Fig. 7 | CCN2 impairs *Gli1* expression and proliferation in chondrocytes, and is downregulated in *Pan-Cart-p21^MOE* mice. a** Pseudobulk Differential Gene Expression analysis of scRNA-seq data (seen as a heatmap) using edgeR. Representative images (**b**) and quantification (**b'**) of *Ccn2* expression (visualised by HCR). PZ, RZ, GoR: proliferative zone, resting zone, groove of Ranvier (separated from cartilage by dashed line). *n* = 5 Ctl, 5 *Pan-Cart-p21^MOE*. Quantification of *Ccn2* expression in Gli1+ (**c**) and Gli1− (**c'**) chondrocytes (*n* = 4 Ctl, 5 Exp). Ex vivo culture method (**d**, *Col2a1-rtTA* only present in EXP embryos) and representative images of *Gli1* or Ki67 staining (**e**); quantified in the indicated zones (**d'**, **e'**). *n* = 7 CTL, 5 EXP. Boxed regions are shown magnified on the sides. **f** Proposed model of how CCN2 is produced in specific cartilage zones limits *Gli1* expression in Pdgfra-derived chondrocytes during normal growth, whereas reduced CCN2 levels due to impaired growth in *Pan-Cart-p21^MOE* limbs lead to unrestricted *Gli1* activation in these cells, some of which become LLCPs. Note that the model includes the possibility that we have not specifically tested, that the articular region contributes cells that, in normal conditions, would give rise to the articular cartilage, not the growth plate.

**Unilateral Ccn2 misexpression model.** The *Pitx2-Cre* mouse line was crossed with the *Col2a1-tTA* one that we previously described[11], and then inter-crossed to generate *Pitx2-Cre/Cre; Col2a1-tTA* mice. The *Tigre^Dragon-Ccn2* mouse line was previously described[15]. Experimental and control animals were generated by crossing *Pitx2-Cre/Cre; Col2a1-tTA/+* females with *Tigre^Dragon-Ccn2/Dragon-Ccn2* males (i.e. homozygous for the conditional misexpression allele). The separation of control and experimental animals was based on the tTA genotype. Pregnant females were not given Doxycycline to allow for tTA activity from the onset of chondrogenesis.

## Table 1 | Primers used for genotyping of the new targeted lines

| ES1 5'-GT PCR (245 bp) 64 °C | |
|---|---|
| MG36 5'-GT step2 up | CGACCTCTAGAAGTAAGCTTTGAGC |
| MG36 5'-GT step2 down | CCTCCCCTGGCACAACGTAA |
| **ES1 3'-GT PCR (449 bp) 64 °C** | |
| MG36 3'-GT step2 up | ATCTTAGCCAGACGAGCGGG |
| MG36 3'-GT step2 down | AATGACCGCTGTTATGCGGC |
| **TRE-GL GT PCR (448 bp) 62 °C** | |
| TRE-GL GT up | GTGAACCGTCAGATCGCCTG |
| TRE-GL GT down | TGGATCCTTAGCGCTGGTGT |

**Generation of G4 mouse embryonic stem cells with Tigre docking site.** A TIGRE docking site targeting vector (a gift from Hongkui Zeng via Addgene plasmid #61580; http://n2t.net/addgene:61580; RRID:Addgene_61580) was electroporated by Monash Genome Modification Platform into G4 mESCs (a gift from Andras Nagy) to generate an ESC line (MG36) with a docking site in the *Tigre* (*Igs7*) locus, amenable to Flp-mediated cassette exchange. 192 individual clones were picked after 6 days of G418 selection. Out of the 96 analysed clones, 21 were positive by Southern hybridisation with both 5' and 3' probes. After reanimation, these clones were analysed again by Southern hybridisation; 8 positive clones were further characterised by Southern hybridisation using internal neo probe (3 restriction digestions); 7 clones (#9, 13, 17, 32, 56, 61, 62) showed single integration. After sequencing, all positive clones were found to have the correct sequence at F5 and F3 mutant FRT sites. 4 clones were karyotyped: clones #56 and 62 are 100% euploid metaphase, whereas clone#32 was 87% euploid, and clone#9 was 80% euploid.

**Generation of TigreTRE-nucGreenLantern.** AR157 vector, containing a TRE-H2B-GreenLantern and a PGK-Hygro resistance split gene (N-half) flanked by FRT sites, was generated by VectorBuilder (Supplementary Fig. 6a). The vector also contained a Polylox cassette[42] downstream of the GreenLantern, but this feature was not used in this study. This vector was then used by Monash Genome Modification Platform for Flp-mediated cassette exchange (Supplementary Fig. 6a) into clones #56, 62 and 32 of *Igs7*-targeted G4 ES cells (MG36, see above). Thirty micrograms pCAG-FlpO-IRES-Puro + 10 µg AR157, mixed with 0.8 ml of ESC suspension at ~1 × 10^7/ml and transferred to a 4 mm cuvette. After incubation at room temperature for 5 min, electroporated by applying a single pulse (an exponential decay pulse at 250 V, 500 µF capacitance or a square wave pulse at 250 V, 5 ms duration) followed by 5 min at RT, then added into a 10-cm dish precoated with MEFs. Two electroporation (one exponential decay pulse and one square wave pulse) per clone was performed. One day after transfection, transfection efficiency was checked under the fluorescent microscope, and drug selection was started (150 µg/ml hygromycin). At 4 days after transfection, selection was changed to hygromycin 100 µg/ml. At 6 days after transfection, we picked up 4 clones from each parental clone (#56-1 to 4, #62-1 to 4, #32-1 to 4) to MEF-coated 24-well plates. These clones were genotyped by 5'-GT PCR, 3'-GT PCR and TRE-GL specific PCR (see Table 1). All clones were correctly targeted (Supplementary Fig. 6b). Clones #62-1, 62-2, 56-1 and 56-2 were used for two sessions each of blastocyst microinjection. Males ET6 (clone 62-1, 85% chimera), ET28 (clone 62-2, 85% chimera), ET51 (clone #56-1, 80% chimera), ET59 (clone #56-2, 80% chimera) were selected for breeding to the next generation. One of these lines was crossed with *Gli1^CreER* animals to establish the *Gli1^CreER*; *Tigre^TRE-nGL* lines. Six additional highly chimeric males were kept as back-up breeders.

## Single-nuclei and single-cell RNA-seq and analysis

**Single-nuclei RNA-sequencing of the unilateral p21 model.** *Pitx2-Cre/Cre*; *Col2a1-rtTA/+*; *Col2a1-eCFP/+* females were crossed with *Tigre^Dragon-p21/Dragon-p21* males and given Dox (1 mg/ml in drinking water with 0.5% sucrose) at E12.5. The knee region (obtained via orthogonal cuts from the hypertrophic region of the distal femur to the hypertrophic region of the proximal tibia in skinned hindlimbs) of E16.5 Left and Right limbs from uCtl and *Left-Cart-p21^MOE* samples (2 embryos of each genotype) were collected in DMEM with 10% DMSO and 5% calf serum, stored in a MrFrosty (ThermoFisher) at 4 °C for 10 min and then transferred to −80 °C overnight. Tissue lysis and nuclei isolation were done as described by L.G.M. in https://www.protocols.io/view/frankenstein-protocol-for-nuclei-isolation-from-f-5jyl8nx98l2w/v2, in two batches of four samples each (left and right knees from one control and one experimental embryo). Once ~5000 nuclei per sample were collected, we proceeded immediately with the 10x Genomics Single Cell Protocol (https://www.10xgenomics.com/support/single-cell-gene-expression/documentation/steps/library-prep/chromium-single-cell-3-reagent-kits-user-guide-v-3-1-chemistry), minimising the time between nuclei preparation/sorting and chip loading. One chip per batch was used (4 channels in each case, to load left and right samples from one control and one experimental embryo). The libraries were sequenced twice on Novaseq (800 million clusters/read pairs each time, which, when combined, amounted to ~40,000 reads per nucleus across the 8 samples).

Following sequencing, fastq reads were mapped against a custom reference that combined Mus musculus (GRCm38) with transgenic genes (tdTomato, ECFP, and WPRE) using CellRanger (v7.1.0). The resulting aligned reads were analysed via R (4.0.2). Seurat (v.4.2.1) and R/Bioconductor SingleCellExperiment packages were used to generate count matrices. For each sample, the mean of unique molecular identifiers (UMIs) (nCounts) was calculated and used as a threshold to exclude empty droplets. Filtering was manually done according to the nCount and the number of genes. DoubletFinder (v2.0.3) was used to remove the expected doublets in each sample. Harmony R package (v 0.1.0) was also used to mitigate batch effects, and subsequently, the data were consolidated into a single Seurat object.

To identify the appropriate resolution, Clustree v 0.5.0 and Cluster Correlation analysis were used. Visual representation of cluster assignments for individual cells was achieved within a reduced-dimensional space using UMAP. To pinpoint the differentiation marker genes in the specific cluster as compared to the remaining clusters, the Wilcoxon Rank-Sum Test was used. This analysis was conducted using the 'FindAllMarkers' function within Seurat (v.4.2.1), *p* value < 0.05 and only.pos = T, and a log-fold change threshold of 0.1. To identify the cell type annotation, the clusters were mapped with published mouse single-cell hindlimb data[43]. In addition, previously reported cellular biomarkers and Gene Ontology (GO) annotations were utilised to annotate the cell clusters.

## Single Cell RNA-sequencing

**Sample preparation, collection, and sequencing.** Female *Gli1^CreER/CreER* mouse was crossed with *Col2a1-rtTA; Tigre^TRE-p21/TRE-p21*; *R26^LSL-tdT/LSL-tdT* male. At E12.5, the pregnant mouse was administered with doxycycline in drinking water (1 mg/ml with 0.5% sucrose) and food, and Tamoxifen was orally gavaged (120 µg/g) at E13.5. At E14.5, the embryos were carefully dissected. Initially, the sex and rtTA genotyping were determined using PCR-based quick genotyping and hindlimbs were then meticulously dissected.

Afterwards, the tissue underwent treatment with Accumax on a rocker at 31 °C for 25 min (Supplementary Fig. 3a) and every 10 min, the samples were gently shaken. The supernatant was carefully transferred into a tube and placed on ice for preservation, while the cartilaginous portions were exposed to Collagenase B (5 mg/ml; Sigma) for

~90 min at 37 °C in a water bath (pipetted every 20 min). To stop collagenase activity, we then added 700 μl of cold DMEM + 20% FBS for each tube. We then incorporated outer and inner supernatants, and the samples were resuspended in 20 volumes of 1x RBC lysis buffer (including $NH_4Cl$, $NaHCO_3$, and EDTA) and incubated for 10 min at room temperature on a rocker. The cells were then centrifuged at 500 g for 5 min at 4 °C and the cell pellet was then suspended in 900 μl of PBS + 0.05% BSA. After re-washing, the cells were resuspended in 400 μl of PBS + 0.05% BSA. Lastly, the cell suspension was filtered through 40-μm Flowmi® Cell Strainers and cells were stained with DAPI (0.05 μg/ml) and transferred into Falcon™ Round-Bottom Polypropylene (Fisher Scientific). The tdT⁺ and tdT⁻ cells were sorted using flow cytometry (nozzle size of 100 μm, operating at a PSI of 20 and a BOP of 223).

In total, two experimental males (EM), two control males (CM), two control females (CF), and only one experimental female (EF) were used. The cells were counted by an automated cell counter (Thermo-Fisher) using trypan blue staining and all samples exhibited >90% cell viability. Then, the cell suspensions were centrifuged at 500 g for 5 min at 4 °C to reduce the final volume. The samples (S) were then multiplexed based on <u>sex and conditions.</u> Specifically, for S1, EM^tdt+ (Experimental Male) and CF^tdt+ (Control Female) cells were multiplexed, totalling 7500 cells. For S2, EM^tdt- and CF^tdt- cells were multiplexed, amounting to 25,000 cells (at a concentration of 1000 cells/μl with a volume of 25 μl). In S3, EF^tdt+ and CM^tdt+ cells were multiplexed, totalling 5700 cells. Finally, in S4, EF^tdt- and CM^tdt- cells were multiplexed, comprising 24,000 cells. Subsequent steps followed the 10X Genomics protocol (https://www.10xgenomics.com/support/single-cell-gene-expression/documentation/steps/sample-prep/single-cell-protocols-cell-preparation-guide).

The libraries were prepared using Chromium Next GEM Single Cell 3′ Reagent Kits v3.1 (10X Genomics) according to the manufacturer's directions and then sequenced on Illumina (~833 million clusters/read pairs each time) with a length of 100 bp. All cDNA libraries passed quality control thresholds (refer to the GSE File).

**scRNA-seq preprocessing.** The fastq files were aligned against a custom reference comprising *Mus musculus* (GRCm38; obtained from 10X Genomics) along with transgenic genes (tdTom, CreERT2, and WPRE) using CellRanger (v7.2.0). In total, more than 98.2% of Reads Mapped to Genome and 25,190 genes were detected, as well as transgenes. To remove ambient RNA contamination, CellBender V 0.3.0 was used. The expected number of genuine cells and total droplets were estimated based on the CellRanger output, using the Raw_feature_bc_matrix.h5 file as input. Identified UMIs were then mapped and subset from the Seurat object generated by the filtered feature barcode matrix file. DoubletFinder (v2.0.3)[44] was then used to remove expected doublets in each sample, with pK calculated separately for each sample. Then the cells with low nCount_RNA and nFeature_RNA were filtered out.

**Testing for differences in cell type proportions.** To analyse the distinct cellular compositions among conditions, the propeller test available in the speckle R (v.0.0.1, accessible at https://github.com/Oshlack/speckle) was employed. The groups were classified with a false discovery rate (FDR) of ≤0.1 as indicative of notable variations in cell types.

**MultiNicheNet.** To investigate cell-to-cell communication, MultipleNichenet (version 1.0.0)[30] was used. The Seurat object was first converted to a SingleCellExperiment (SCE), and all genes were mapped using the alias_to_symbol_SCE function to 'mouse'. A cutoff point of min_cells = 10 was used, and differentially expressed genes were calculated using the get_DE_info function. The following parameters were applied: logFC_threshold = 0.5, p_val_threshold = 0.05,

fraction_cutoff = 0.05, and empirical_pval = TRUE. To achieve more robust results and minimise cluster dropouts caused by a low number of cells, some subclusters were merged. Ligand-receptor interactions were identified via the 'multi_nichenet_analysis' function, and the results were visualised using MultiNicheNet's built-in functions[30].

**SCENIC analysis.** SCENIC (v1.2.4)[12] was used to assess gene regulatory network activities in each cell population. For snRNA-seq data, pySCENIC[45] was employed and outputs were manually imported into SCENIC in R. A co-expression network was generated using the GENIE3 function, and the Area Under the Curve (AUC) was calculated with AUCell. The AUC values were normalised to a range of 0 to 1 for each regulon and regulon scores were obtained using the 'get_AUC' function. To assess regulon activity changes, *t*-tests and Wilcoxon tests were performed within each cluster, and one-way ANOVA was used to evaluate the regulon activity across conditions.

**MiloR.** To identify differential cell states, the Seurat object was subset to include only Gli1⁺ cells and split by 'ETP' and 'CTP' status, and then we used the miloR (https://github.com/MarioniLab/miloR) package (v1.6.0)[46]. A k-nearest neighbour graph ($k = 30$, $d = 20$) was constructed based on PCA-reduced dimensions. Neighbourhoods were generated, and the number of cells in each neighbourhood was counted. A design matrix incorporating experimental conditions, cell lineage, and sample metadata was created for differential analysis. Neighbourhood distances were calculated using PCA reductions and log-transformed count data. Differential analysis of these distances was performed with testNhoods, and the results were visualised based on SpatialFDR values. UMAP-based dimensionality reduction was visualised using the plotReducedDim function, with cells coloured by ETP and CTP and labelled by cell type.

**Pseudobulk approach**

We used pseudobulk approaches to ascertain gene expression profiles across three distinct conditions in snRNA-seq and in single-cell RNA-seq. Gene expression counts were aggregated within each combination of individuals, and the linear modelling approach from the Limma R package (v.3.64.0) was employed to incorporate our experimental design into the differential analysis. The pseudobulk gene expression matrix was generated using the Seurat2PB function, specifying 'sample' and 'type' parameters to define the sample identifiers and cluster types. The TMM method in edgeR (v.4.6.1) was used to compute normalisation factors. Differential gene identification was performed with a significance threshold of raw $p < 0.05$. Heatmaps were generated to visualise differential expressions, using the pheatmap (v.1.0.12) in R (4.0.2).

**EdU/CldU incorporation and detection**

A solution of 5-Ethynyl-2′-deoxyuridine (EdU) was prepared at 6 mg/ml in phosphate-buffered saline (PBS). This solution was administered at a dose of 60 μg/g, subcutaneously (s.c.) for pups and 30 μg/g intraperitoneally (i.p.) for pregnant females, 1.5 h before euthanizing the mice (or several days earlier for pulse-chase experiments). To detect EdU, a click chemistry reaction with fluorescein-conjugated azide (Lumiprobe #A4130) was performed once the immunohistochemistry and/or TUNEL staining were completed on the same slides. Briefly, the working solution was prepared in PBS, adding $CuSO_4$ (Sigma # C1297) to 4 mM, the azide to 0.4 μM and freshly-dissolved ascorbic acid (Sigma #A0278) to 20 mg/ml, and incubating the sections for 15 min at room temperature in the dark, followed by PBS washes. When used, 5-chloro-2′-deoxyuridine (CldU) was prepared at 6 mg/ml in PBS, and injected s.c. to pups (60 μg/g), 1.5 h before euthanising the pups. To detect both EdU and CldU, after the first EdU click reaction (as above), a second click reaction was performed with a different azide: azido-methyl phenyl sulphide (AMPS, 2 mM final concentration) to block access to EdU alkyne. Sections were incubated for 15 min at room

temperature in the dark, followed by 1x PBS wash and 1x PBSTx (0.1%) wash. Then, sections were blocked with 5% goat serum in PBSTx 0.1% for 5 min at room temperature and incubated with anti-CldU 1/500 (rat monoclonal BU1/75 (ICR1), Abcam ab6326) in 5% goat serum in PBSTx 0.1% for 2 h at room temperature, followed by PBSTx (0.1%) washes. Anti-rat secondary antibody diluted 1/500 in PBS with DAPI was incubated for 1 h at room temperature. The sections were washed by PBSTx (0.1%) and mounted with Fluoromount.

## Sample collection and histological processing

Mouse embryos were euthanized using hypothermia in cold PBS, while mouse pups were euthanized by decapitation. Upon collection of the embryos or pups, the limbs (including full tibiae and/or femora) were carefully dissected out in cold PBS, skinned, and fixed in 4% paraformaldehyde (PFA) for 2 days at 4 °C. Samples of P1 or younger were processed without decalcification. For P3, P5, and P7 samples, decalcification was performed by immersing the specimens in 0.45 M ethylenediaminetetraacetic acid (EDTA) in PBS for 3, 5, and 7 days, respectively, at 4 °C. Following several washes with PBS, the limb tissues were cryoprotected in PBS containing 15% sucrose and then equilibrated in 30% sucrose at 4 °C until they sank. The hindlimbs were then oriented sagittally, facing each other, with the tibiae positioned at the bottom of the block (closest to the blade during sectioning) and embedded in Optimal Cutting Temperature (OCT) compound using cryomolds (Tissue-Tek). The specimens were frozen by immersing them in dry-ice-cold iso-pentane (Sigma). Serial sections with a thickness of 7 μm were collected using a Leica Cryostat on Superfrost slides. The sections were allowed to dry for at least 30 min and stored at −80 °C until further use. Prior to conducting histological techniques, the frozen slides were brought to room temperature in PBS, and the OCT compound was washed away with additional rounds of PBS rinses.

## Ex vivo culture

The culture medium was prepared by mixing 200 μL penicillin/streptomycin (10,000 U/ml) with 48.9 ml high-glucose DMEM, 500 μL Glutamax, 100 mg BSA, 2.5 mg ascorbic acid, and 400 μL β-glycerophosphate, then filtered through a 0.2-μm filter. E14.5 embryos were dissected and transferred to pre-warmed media under a tissue culture hood. Each hindlimb was placed in cold Accumax and then digested at 31 °C for at least 35 min. The samples were then put on ice, and the tibia and femur were dissected individually. Limbs were cultured in 48-well plates with 500 μL of media at 37 °C in a 5% CO$_2$ incubator for approximately 2 h before initiating treatment. For the treatment, human recombinant CCN2 (Peprotech, 17840573) was used. One limb from each pair was treated with 100 ng/ml of CCN2 + Doxycycline, while another set received Doxycycline + Vehicle (1% BSA in Saline). After 2 days, the limbs were treated with 10 μM EdU for 1 h before fixation in PFA.

## Micro-CT and bone length analysis

The micro-CT and bone length analysis was as previously described[47]. Briefly, samples were retained and fixed in 4% PFA as residual tissues from other experiments in the Rosello-Diez lab. Whole femora and humeri were scanned using a Siemens Inveon PET-SPECT-CT small animal scanner in CT modality (Monash Biomedical Imaging). The scanning parameters included a resolution of 20 and 40 μm, 360 projections at 80 kV, 500 μA, 600 ms exposure with a 500 ms settling time between projections. Binning was applied to adjust the resolution to 2 × 2 for 20 μm scans and 4 × 4 for 40 μm scans. The acquired data were reconstructed using a Feldkamp algorithm and converted to DICOM files using Siemens software. For the analysis and bone length measurements, Mimics Research software (v21.0; Materialize, Leuven, Belgium) equipped with the scripting module was utilised to develop the analysis pipeline[47].

## Immunohistochemistry and TUNEL staining

For antigen retrieval, the sections were subjected to citrate buffer (10 mM citric acid, 0.05% Tween 20 [pH 6.0]) at 90 °C for 15 min. Afterward, the sections were cooled down in an ice water bath, washed with PBSTx (PBS containing 0.1% Triton X-100). To perform TUNEL staining, the endogenous biotin was blocked using the Avidin/Biotin blocking kit (Vector #SP-2001) after antigen retrieval. Subsequently, TdT enzyme and Biotin-16-dUTP (Sigma #3333566001 and #11093070910) were used according to the manufacturer's instructions. Biotin-tagged DNA nicks were visualised using Alexa488- or Alexa647-conjugated streptavidin (Molecular Probes, diluted 1/1000) during the incubation with the secondary antibody.

For immunohistochemistry staining, sections were incubated with the primary antibodies prepared in PBS for either 1.5 h at room temperature or overnight at 4 °C (see list of antibodies below). Following PBSTx washes, the sections were incubated with Alexa488-, Alexa555-, and/or Alexa647-conjugated secondary antibodies (Molecular Probes; diluted 1/500 in PBSTx with DAPI) for 1 h at room temperature. After additional PBSTx washes, the slides were mounted using Fluoromount™ Aqueous Mounting Medium (Sigma). The antibodies used, along with their host species, vendors, catalogue numbers, and dilutions, were as follows: mCherry (goat polyclonal, Origene Technologies #AB0040-200, diluted 1/500), BrdU/CldU (rat monoclonal BU1/75 (ICR1)), Abcam #ab6326 diluted 1/500, GFP (chicken polyclonal, Abcam #ab13970, diluted 1/1000), nanoGFP (alpaca, monoclonal, Chromotek #gb2AF488-50 diluted 1/300), Ki67 (rat monoclonal (SolA15), ThermoFisher #14-5698-82 diluted 1/100), mKate2 (camel monoclonal 1H7, Abnova #RAB00798, diluted 1/100), PDGFRA (goat polyclonal, R&D # AF1062, diluted 1/100).

## In situ hybridisation

To perform in situ hybridisation, sections were fixed in 4% PFA for 20 min at room temperature, washed in PBS, and treated with 4 μg/ml Proteinase K for 15 min at 37 °C. After washing in PBS, the sections were refixed with 4% PFA, followed by 10-min treatment at room temperature with 0.25% acetic anhydride (Sigma #320102) freshly added to 0.1 N pH 8 triethanolamine (Sigma #90279). Subsequently, the sections were washed in PBS and water and incubated with pre-hybridization buffer (50% formamide, 5x SSC pH 5.5, 0.1% 3-[(3-Cholamidopropyl)dimethylammonio]-1-propanesulfonate (CHAPS), 0.05 mg/ml yeast tRNA, 0.1% Tween 20, 1x Denhardt's) at 60–65 °C for 30 min. The sections were then incubated with 1 μg/ml preheated riboprobes and subjected to hybridisation at 60–65 °C for 2 h. Post-hybridisation washes were performed using post-hybridisation buffer I (50% formamide, 5x pH 5.5 SSC, 1% SDS) and II (50% formamide, 2x pH 5.5 SSC, 0.2% SDS) preheated at 60 °C for 30 min, respectively. The sections were then washed with maleic acid buffer (MABT: 100 mM maleic acid, 150 mM NaCl, 70 mM NaOH, 0.1% Tween 20) and blocked with 10% goat serum, 1% blocking reagent (Roche #11096176001) in MABT at room temperature for 30 min. Next, the sections were incubated overnight at 4 °C with anti-digoxigenin-AP (Sigma #11093274910) diluted 1/4000 in MABT with 2% goat serum and 1% blocking reagent. After several MABT washes, the sections underwent 2 washes with NTMT/AP buffer (0.1 M Tris-HCL pH 9.5, 0.1 M NaCl, 0.05 M MgCl$_2$, 0.1% Tween 20), with the second one containing 1 mM levamisole and then developed colour using BM purple (Roche #11442074001) at 37 °C. Following a wash in PBS, the sections were fixed for 10 min in 4% PFA, counterstained with Nuclear Fast Red (Sigma #N3020) at room temperature for 10 min, and rinsed in water. Dehydration was performed by passing the sections through 70%, 90% ethanol, absolute ethanol, and xylene. Finally, the sections were mounted with DPX (Sigma #100579).

Most slides were imaged with a Zeiss Imager Z2 equipped with a colour camera. The *Pdgfra* developmental series was imaged with a

Hamamatsu Nanozoomer at 40×. Images were exported with NDPView software.

## Hybridisation chain reaction (HCR)

The slides were thawed for 20 min in RNAse-free PBS (DPBS; GIBCO, Invitrogen, LOT: 866200), then rinsed twice with DPBS and fixed in ice-cold 4% PFA for 15 min at 4 °C. The samples were then dehydrated by being immersed in 50% ethanol, 75% ethanol, and 100% ethanol for 5 min at RT. The slides were washed with DPBS. A total of 200 μl/slide of probe hybridisation buffer was added to the sample and incubated for a minimum of 10 min at 37 °C. After draining the excess buffer, 95 μl/slide of probe solution at a final concentration of 16 nM were added to the slide and the incubation was carried out for at least 12 h.

To remove excess probes, slides were sequentially washed for 15 min each with 75%, 50%, and 25% probe wash buffer (diluted in 5× SSCT), followed by a final wash with 100% 5× SSCT. Following these steps, the slides were immersed in 100% 5x SSCT for 5 min at room temperature. The amplification stage was initiated by adding 200 μl/slide of amplification buffer on top of the sample, and the samples were pre-amplified in a humidified chamber for 30 min at room temperature. Meanwhile, 6 pmol of hairpin h1 and 6 pmol of hairpin h2 were separately prepared by snap cooling 2 μl of a 3 μM stock (heated at 95 °C for 90 s and cooled to room temperature in a dark drawer for 30 min). Each hairpin was added to 100 μl/slide of amplification buffer at room temperature. After removing the pre-amplification solution from the slides, 98 μL of the hairpin solution was added on top of the samples, and incubation was carried out for a minimum of 12 h in a dark, humidified chamber at room temperature. Subsequently, the slides were subjected to twice washing with 5x SSCT at room temperature for 30 min. Finally, the samples were washed for 5 min with 5x SSCT at room temperature. Following these, the slides were allowed to dry and then mounted using antifade mounting reagent (Fluoromount-G™ Mounting Medium, Cat: 00495802, USA).

## Imaging

Sagittal sections of the limbs were captured, with a focus on the area between the distal femora and proximal tibiae. Typically, at least two sections per limb were analysed, although in most cases, four sections were examined. In the case of cultured distal femora, frontal sections were used as they provided better identification of the different epiphyseal regions. To determine the boundaries of the resting zone (RZ), proliferative zone (PZ) and hypertrophic zone (HZ), morphological criteria were applied. The RZ was considered to include the articular region (except the outer-most two cell layers) at the stages before a SOC exists. The transition between round (resting) and flat (columnar) nuclei, forming an arch along the upper point of the grooves of Ranvier, was considered as the start of the PZ. On the other hand, the transition towards larger, more spaced nuclei (pre-hypertrophic) marked the end of the PZ. The point where the pericellular matrix exhibited a sharp reduction around enlarging chondrocytes was designated as the beginning of the HZ. The distal end of the last intact chondrocyte served as the endpoint of the HZ. For imaging, brightfield and fluorescence images were acquired using a Zeiss inverted microscope (Imager.Z1 or Z2) equipped with Axiovision software (Zeiss/ZenBlue). Mosaic pictures were automatically generated by assembling individual tiles captured at 10× magnification for brightfield images or 20× magnification for fluorescence images.

## Image analysis and quantification

The regions of interest, such as resting, proliferative and hypertrophic zones (RZ, PZ, HZ) and the SOC, were identified from imaged sections of the multiple models (left and right; experimental and control proximal tibial cartilage). Consistent parameters such as brightness, contrast, filters and layers were kept the same for all the images in the same study.

## Cell-count analysis

The RZ and PZ were identified and segmented from sections stained for DAPI, tdTomato, and TUNEL or EdU using macros in FIJI. Details of the macros are available upon request. The number of cells in the region of interest (RZ & PZ) were measured using CellProfiler. Since in our hands, CellProfiler did not work properly with more than 3 channels, a pipeline was generated to split the four channels into DAPI on one hand and the 3 other channels on the other hand. The two sets of images were run on CellProfiler separately. Threshold bounds were manually selected with an initial set of images, and kept approximately constant for the rest. However, manual curation was applied to every single image by checking whether the outlines of the automatically identified objects matched the eye of the curator, who was blinded to the genotype. Threshold fine-tuning was applied if necessary. An example CellProfiler pipeline has been uploaded as Supplementary material.

## Statistical analysis

Statistical comparisons were performed using appropriate tests based on the experimental design. An unpaired $t$-test was used for comparisons involving one variable and two conditions. One- or two-way ANOVA was utilised for comparisons involving one or two variables, respectively, and two or more conditions. The Shapiro-Wilk test was used to confirm normality of the data when in doubt. All statistical analyses were two-sided and conducted using Prism 10 software (GraphPad).

## Reporting summary

Further information on research design is available in the Nature Portfolio Reporting Summary linked to this article.

## Data availability

The single-nuclei and single-cell RNA-seq datasets have been deposited at the NCBI Sequence Read Archive (SRA) with accession numbers PRJNA1136579 and PRJNA1138445, respectively. These can be accessed via the Gene Expression Omnibus (GEO), with accession numbers GSE273540 and GSE273538, respectively. Source data are provided with this paper.

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

## Acknowledgements

The authors acknowledge the Monash Genomics and Bioinformatics Platform for their excellent technical help, as well as the Monash Genome Modification Platform, for their help with gene targeting and for assistance with designing and executing the genotyping strategy. The authors thank Timothy Semple for his help with library preparation for the snRNA-seq dataset and Ricky Johnstone for providing access to bioinformatics expertise. The authors also thank anonymous reviewers for their insightful comments. This work was supported by HFSP CDA00021/2019-C (to A.R.-D.) and NHMRC Ideas grant 2002084 (to A.R.-D. and C.H.H.). The Australian Regenerative Medicine Institute is supported by grants from the State Government of Victoria and the

Australian Government. The Novo Nordisk Foundation Center for Stem Cell Medicine, reNEW, is supported by a Novo Nordisk Foundation grant number NNF21CC0073729.

## Author contributions

X.Q.: data acquisition and analysis, supervision; E.R.: data acquisition and analysis, figure preparation, manuscript editing; A.K.C.: data acquisition and analysis, figure preparation; S.L.A.: data analysis, manuscript editing, supervision; K.K.V.: data acquisition and analysis, figure preparation; F.J.R.: data analysis; M.Z.: data analysis; L.G.M.: experimental design, data acquisition; C.H.H.: data analysis, manuscript editing; D.R.P.: tool generation, data analysis, supervision; A.R.-D.: conceptualisation, experimental design, data analysis, funding acquisition, supervision, figure preparation, manuscript drafting.

## Competing interests

F.J.R. receives institutional support as a coinvestigator and is subcontracted by the Peter MacCallum Cancer Centre for an investigator-initiated trial, which receives funding from Sanofi/Regeneron Pharmaceuticals. All other authors declare no competing interests.
