## [Transparent Peer Review file · Nature Communications]

Gli1-expressing stromal cells are highly reparative precursors of long-lived chondroprogenitors in the fetal murine limb

Corresponding Author: Dr Alberto Rosello-Diez

Version 0:

Reviewer comments:

Reviewer #1

(Remarks to the Author)

In this manuscript, Qu et al. performed various transgenic mouse experiments to understand the origin and nature of skeletal stem cells that contribute to renewal of growth plate chondrocytes during postnatal and adolescent skeletal growth. The authors studied long-lived chondroprogenitors by performing Gli1-lineage experiments, and also used a sophisticated mouse model in which p21-driven cell-cycle arrest was challenged in the prenatal growth plate to examine the compensatory (reparative) mechanisms. Based on the data presented in this manuscript, the authors draw several conclusions: (1) Prenatal Gli1-lineage cells play a significant role as long-lived progenitors in postnatal growth plate renewal and function; (2) stimulation of cell-cycle arrest increases the number of long-lived Gli1-lineage progenitors, some of which are recruited from surrounding tissues in the growth plate; (3) Pdgfra-lineage cells, which likely reside in the perichondrium, contribute to the addition of Gli1+ lineage progenitors into the growth plate; and (4) Compensatory mechanisms by Gli1+ cells involve de novo expression of Gli1 and Ccn2 down-regulation that facilitates Gli1 expression. The strengths of this manuscript include providing clear evidence that prenatal cartilage cells significantly contribute to maintaining growth plate function over an extended period of time until skeletal growth is nearly complete in mice, a new and intriguing finding that Gli1+ cells play an important role in recovery of the growth plate function from cell-cycle arrest-induced damage, and presenting a novel reporter-retention model to visualize the cell dynamics of chondroprogenitors in the growth plate.

On the other hand, one of major weaknesses is that conclusions 3 and 4 are not convincingly supported by the data provided. Although clarifying the origin of chondroprogenitor cells listed in conclusion 3 is very interesting and worth sharing with readers, this conclusion is not agreeable due to serious concerns, including insufficient validation of the effective period and recombination efficiency of DOX and tamoxifen, leading to uncertainty in the experiment systems, and an absence of results on tempo-spatial changes in Gli1 and Pdgfr2 transcripts or proteins. There are additional concerns on the data presentation and interpretation, and experiment designs and methods.

In summary, the chondroprogenitors have been thought to be present in prenatal or postnatal cartilage primordium, but also to arise outside the cartilage, such as the perichondrium and the groove of Ranvier. This manuscript clearly demonstrates the presence of some long-lived chondroprogenitors in prenatal cartilage, which is the originality and significance of this study. Although this study also aimed to answer the long-standing and important questions of whether and how chondroprogenitors are supplied from tissues outside the growth plate, the results provided are inconclusive. Therefore, in its current form, this manuscript unlikely provides significant advances in our understanding of growth plate chondroprogenitors.

Major specific comments

1. Clarification of transgenic mouse strains. Most methods for generating compound transgenic mice have been well described. However, combining different strains in compound transgenic mice may introduce biases depending on the nature of the mouse strains. Clarify the mouse strains of all transgenic mice used in this study, and state how the strategy of transgenic mouse generation minimizes such experiment flaws. Gli1CreER mice have been made by insertion of CreER into the Gli1 locus at the 1st exon, and Southern blot analysis has shown a shorter size of Gli1 recombined allele (Cell. 118, 505-516, 2004). Confirm that the CreER insertion does not disturb endogenous Gli1 expression and the use of homozygous Gli1CreER mice for breeding is appropriate.

2. The study uses two types of cell-cycle arrest models, Left-Cart-p21 and Pan-Cart-p21. The phenotypes of these two

models are different regarding the left-right discrepancy and completion of catch-up. The Left-Crt-p21 shows growth retardation (Ext.Fig.2c) while the Pan-Cart-p21 growth plate seems to catch up completely (Fig.1b'). The compensatory mechanism for cell-cycle arrest in these two models may be similar to a greater or lesser extent, but may differ from each other. Combining the results of the two models is therefore complex and difficult to interpret accurately. As the current study starts from previous results obtained from Left-Cart-p21, the authors could have included the results shown in Figure 1a, but in the remainder of the manuscript it would be better to focus exclusively on the Pan-Cart-p21 model.

3. This study largely relies on the lineage tracing. To clarify whether the increase in the number of Gli1+ and Pdgfra+ cells is due to de novo Gli1/Pdgfra expression, an expansion of lineage cells or both, and where these de novo expressing cells reside, it is necessary to examine tempo-spatial changes in gene expression of Gli1/Pdgfra in the lineage-tracing models. The authors include Gli1 expression in Pdgfra-lineage cells and vice versa (Fig.6b,d), and similarly, Gli1 expression in Gli1-lineage tracing model and Pdgfra expression in Pdgfra model should be included. It would be also informative to show the relation among p21 expressing cells, Ki67-cells, Ccn2 expressing cells and Gli1-expressing and lineage cells to understand which cells respond to cell-cycle arrest.

4. Fig. 1c, c' and Ext Fig 4. There is no evidence that tamoxifen clearance is within one day. It is likely that tamoxifen remains active in the body for more than one day. At least Tamoxifen should be injected in prior to Dox treatment. The data do not support the statement "This result suggests that while the existing population of Gli1+ cells contributes to the repair, some Gli1- cells become Gli1+ after p21 induction in the cartilage" (lane 143-144). As discussed above, it is more convincing to address upregulation of de novo Gli1 expression by demonstrating upregulation of Gli1 transcripts in non-Gli1+ cells.

5. Fig. 2. Gli1-Cart-DTA group is needed as additional control. As shown in Fig 5b', growth plate function in the Gli-Cart-DTA mice was seriously affected. DTA ablation would affect not only compensated Gli1+ cells but also other Gli1+ progenitors present in the growth plate, and Fig 2 experiments may not demonstrate a specific role for Gli1+ cells in compensating cell-cycle arrest. "This result demonstrates that Gli1-derived chondrocytes are required to compensate for cell-cycle arrest in the fetal cartilage" (lane 162-164) seems overestimated.

6. Fig. 4. The authors show that most GL retaining cells were Gli1-lineage, indicating that Gli-lineage cells contain LLCs. They also show that the GL retaining cells decreased in the Gli1-lineage in the RZ, and conclude that GL-unlabeled Gli1-lineage progenitors are newly added Gli1-lineage cells from outside the growth plate. However, to reach this conclusion, there are at least two different interpretations that must be ruled out.

(1) Extended Fig. 6C shows that the cells at the top of the cartilage did not express GL, but Gli-lineage cells (Tom+), suggesting that even in the growth plate, some Tom+ cells are not GL-labeled, especially in the RZ. If these non-labeled LLCs produce progenitors in the RZ, the GL-retaining population in the Gli1-lineage should decrease.

(2) Once the Gli1-lineage LLCs were labelled by GL, and undergo a certain number of asymmetric divisions to produce progeny, their GFP intensity may decline beyond detection at P21 or later, and only quiescent GL-LLCs may retain GL until P21. Therefore, the reduced GL-retention in Gli1+ lineage cells could be due to dilution of the label by cell division, rather than "newly added" Gli1 lineage LLCs.

7. Pdgfra-related experiments have serious concerns regarding the statement "we chose platelet derived growth factor alpha (Pdgfra) as a marker of stromal cells that is not expressed in the cartilage" (lane 245-246). This reviewer is concerned that, judging from the data given, the Pdgfra lineage cells do not necessarily come from outside the growth plate, and that Pdgfra-related experiments may not be of significant relevance in this manuscript.

Some chondrocytes are clearly positive to Pdgfra in Ext Fig. 5b. Ext Fig. 5b' image (likely Pdgfra-GFP reporter) shows that some cells look positive when the image brightness is increased a little bit. To support the statement "the surrounding tissues can contribute to cartilage" (lane 250-251), it is necessary to provide the results that no Pdgfra+ cell is present in cartilage in the Pdgfra;R26LSL-tdT mice at E13.5 after Dox injection at E12.5. Higher resolution (higher quality) of in situ hybridization image of Pdgfra from E12.5 to E14.5 should be provided to prove that Pdgfra expression is exclusive in tissues outside the cartilage. Fig. 6a shows Pdgfra-TdTom positive cells are more than 30% in the control cartilage at E14.5. In Fig.6d', in the control group, PDGFRA+ cells were observed in the growth plate at E14.5. The data are open to several interpretations: 30% of chondrocytes are added from the Pdgfra+ cells in tissues outside the cartilage, Pdgfra expression is induced inside the cartilage, Pdgfra+ cells are present in cartilage already, or a mixture of these events occurs.

Minor specific comments

1. "fetal-cartilage targeted cell-cycle arrest" in the abstract is difficult to understand without reading the main text.

2. Fig 1a'. "(GLI1) transcription factor was significantly upregulated in the left experimental (ExpL) osteochondroprogenitors and to a lesser extent in resting chondrocytes". In Fig.1a', it seems that Gli1 expression is more prominent in resting chondrocytes. To demonstrate the difference between control and experimental groups, other visualization method such as fold change, or a split violin plot would be better.

3. Fig 1c'. The statement "in control conditions, most chondrocytes derived from fetal Gli1+ cells are only transient and eventually eliminated from the cartilage"(lane 127,128) is somehow incorrect because Fig.1c' shows a robust population of Gli lineage cells stayed in the cartilage at P0 compared to E14.5.

4. Fig 1c, c'. The trabecular area should not be included in the "outside" population because there should be a substantial lineage+ population which are derived from the Gli1 lineage chondrocytes (i.e. transdifferentiation).

5. Fig.3. "short-lived clones that get depleted over time, and long-lived ones that remain dormant at peri-natal stages and

become more active after the second postnatal week” (lane188-190). It is possible that the same Gli1 lineage progenitors give rise to short columns at earlier time points but start to produce long columns at the second week. There is no convincing evidence showing that the short-lived progenitors were depleted. There are a certain number of GLI1+ cells in RZ without making column. How do the authors interpret them? This population seems quiescent, and would be GL retaining cells in the Gli1-lineages cells in the RZ as described in the major comment 6 above.

6. Chondrocytes generated in the last postnatal week from Gli1+ cells are required for long-term bone growth (lane 231-232). The paragraph is not consistent with this section title.

7. Fig.6b', c. These figures clearly show that the percentage of Pdgfra-lineage cells in Gli1-expressing chondrocytes decreased upon p21 induction, and there was a robust expansion of Gli1+ cells within the growth plate. These results suggest that some Gli+ cells are not derived from the Pdgfra lineage. To validate the significant contribution of Pdgfra lineage to the compensatory mechanism, the authors might perform DTA experiment in these cells.

8. Fig 7e. If the authors would like to argue the involvement of Ccn2 in addition of Gli1/Pdgfr2a+ cells into the growth plate, they need to provide evidence that down-regulation of Ccn2 stimulates or up-regulation of Ccn2 inhibits cell migration from the perichondrium (groove of Ranvier) in ex vivo or in vivo.

9. Discussion. Previous reports (Nature 2013 499:491-5 and J Bone Miner Res 2019, 34:964-974) have performed the lineage tracing experiments with CtskCre mice and shown that a only a small proportion of cells in the groove of Ranvier constitute the growth plate, but contribute little to making columns of the growth plate. The authors should discuss these findings.

Reviewer #2

(Remarks to the Author)

In this study, Qu and colleagues ask how embryonic limbs are able to compensate their growth when injured by a loss of proliferative cells. Using sophisticated genetic mouse models that enable mosaic cell cycle arrest in the left limb only, they show that descendants of Gli1-expressing cells are necessary for this compensation, and in the absence of compensation serve as an important progenitor cell population. Fetal Gli1+ cells are a population of label retaining progenitor cells that give rise to chondrocytes during postnatal bone development and are necessary for normal bone length. Upon injury, Pdgfra-expressing cells give rise to Gli1+ expressing cells, identifying a mechanism for amplification of this population to compensate for growth in the injury model. Finally, they find that CCN2, a negative regulator of bone growth, is inhibited in Cart-p21MOE experimental samples, suggesting a possible mechanism for the expansion of Gli1+ cells.

Overall, this is an exciting and novel study. The experimental design has multiple strengths, including sophisticated, novel genetic models, rigorous experimental controls and well-designed experiments with appropriate replicates and quantification. The findings of a fetal progenitor cell population are novel and will serve as a foundational study for understanding how this progenitor population is regulated both during normal bone development as well as in growth compensation. Although I do not have any major concerns, several minor concerns are listed below for the author's consideration.

1. Although the Shi et al, 2017 study implicating Gli1+ cells in fracture repair is cited, additional work from Francesca Mariani's lab has also shown that Hedgehog signaling is essential for fracture repair, additional discussion of this is warranted in the discussion. For example, the Mariani lab has found that Hedgehog signaling is important in a rib fracture model Kuwahara et al, 2019 Elife8, e40715. Please consider a brief addition to the discussion that considers these findings with the newly defined Gli1+ fetal progenitor cells in this study.

2. Consider replacing Fig 1a (which on its own is confusing) with Extended data Fig 2a, which greatly clarifies the experimental foundation.

3. Figure 1a'. The enrichment of Gli1 activity by SCENIC is confusing in its presentation. Are the depicted regulons shown in order of enrichment – and by what criteria? If not, how are they picked? Is this tested using experimental only (left versus right) is this a multiparametric analysis? Are all pairs tested for significance with only Gli1 reaching significance (in the outlined and dashed domains) or are other domains significantly enriched as well? Please also clarify the abbreviations used in the figure panel so that they are consistent between what is described in the text and on the figure image (e.g. 'Expl' vs 'EL').

In addition to GLI1 regulon activity, were Gli1 or Gli1-regulated network genes differentially expressed between the left and right knees of the Left-Cart-p21MOE mice? Even though the subsequent data makes a compelling case that Gli1+ cells provide compensation in growth-challenged limbs, this first evidence seems tenuous as presented.

4. Figure 7A. Does this show the most significant DEGs (as measured by FC/FDR?) or is this a curated subset? If the latter, please provide a statement about where Ccnd2 fits into this overall set of DEGs.

5. Figure 7: Although there is interesting data presented on Ccn2, Ctnd2 (delta catenin) appears highly upregulated in the experimental Cart-p21MOE samples compared to controls. Please consider commenting on this and how it might fit into models for Gli-lineage activation.

6. Figure 7C: I understand this figure from the description in the text but I'm confused about the label 'Asmu-Swiss'. Isn't this the Pan-Cart-p21MOE model?

7. Methods section lines 685-700 what seems to be a redundant, less detailed description of the HCR protocol that should be deleted.

8. Extended data. With the exception of Extended Data Fig. 5, it was a bit challenging to figure out which figure was which since they aren't numbered and don't have embedded legends. Consider adding Figure numbers and, if allowable by the journal, embedded figure legends.

Reviewer #4

(Remarks to the Author)

Version 1:

Reviewer comments:

Reviewer #1

(Remarks to the Author)

The revised manuscript has been improved by addressing the concerns. However, several important points remain to be clarified. There are also additional suggestions.

1. 1. Transgenic mouse strains. Please provide information about the background strains of transgenic mice used by the authors for breeding. Mixing background strains will result in variability in growth and characteristics. Background information is important for scientific rigor and will be useful for future experiments.

2. The origin of Gli1 and Pdgfra-lineage cells.

Extended Data Figure 5c. This reviewer agreed that a majority of Pdgfra transcripts and Pdgfra-lineage cells were outside the cartilage where the authors marked by the white-dotted line. However, Pdgfra transcripts (white) and Pdgfra-lineage cells (weak red signals) were also found inside the white line and likely double-labeled with GL (green). The authors should not ignore these cells. Therefore, the possibility remains that some Pdgfra-expressing and -lineages cells may exist inside the cartilage. Note that this reviewer does not mean to deny the author's idea that Gli1- and Pdgfra-lineage cells are added outside the cartilage and that challenging increased number of Gli1-/Pdgfra-lineage cells. Indeed, addition of chondrocytes from the surrounding tissue such as Ranvier's groove has been indicated previously.

Extended Data Figure 6d and L.266-268 "As shown in Ext. Data Fig. 6d, we could not detect almost any tdTom+ nGL- cells within the Matn1+ region, ruling out that possibility." --- This statement is not agreeable at this moment. Please provide gray scale single raw images. It is difficult to monitor exact positive signals of composed images. It seems that Matn1 transcripts (Magenta) and nGL-cells exist outside/above the white dotted line. The nGL system and Matn1 expression may label broader areas than the authors define as cartilage.

3. Tamoxifen and Dox efficiency and clearance

L.146-148, ", with TM clearance mostly taking place before any potential de novo Gli1 expression. Indeed, it was previously shown with a cartilage-driven CreER that recombination is complete 24h after TM injection¹⁹". - Reference 19 shows that TM is sufficient to induce Cre recombination within 24 hours by monitoring LacZ expression, but does not show how long it takes for TM to be cleared.

Figure 4c. The extension of the Dox administration period increased the number of tdTom-negative/nGL-positive cells in the RZ, suggesting that the extension of the Dox administration period may increase the number of labeled cells even inside and the cartilage. --- The experiment in Figure 4 is based on the idea that 24 hours of drug administration fully activates cre recombination and drug clearance is achieved within 24 hours. Although high efficiency of tamoxifen administration is demonstrated, the mouse system used in this study is not exactly the same as the study cited by the authors, and Dox efficiency and clearance are not evident. If one day of Dox administration does not achieve full efficiency and several days of exposure are required, a simple interpretation of the data as the authors did would be not made. The authors need to consider more carefully the uncertainties in clearance and efficiency of TM and DOX in in vivo experiments.

4. Image analysis:

-- The authors' determination of negative/positive signals in images, such as the Extended Data in Figure 6C, raises the question of how to define the signal threshold. Please provide a more detailed method beyond the use of CellProfiler, since cell analysis is one of the most important parameters in this study.

- The authors state that "Non-parametric tests were selected when the assumption of normality could not be met." It is likely that all data (comparison of bone length and cell analysis) are statistically analysed by parametric one way or two way ANOVA. Provide the method how to determine normality.

Other modest to minor comments

- The definition of the resting and proliferating cells in E14.5 and E17.5 tissues is unsure. Looking at the histology shown in Extended Data Figure 7, especially in E14.5 limbs, the difference between resting and proliferating chondrocytes is difficult to be defined by "The transition between round (resting) and flat (columnar) nuclei, forming an arch along the upper point of the grooves of Ranvier, was considered as the start of the PZ". The top border of the RZ is also unsure. The top border of RZ needs to be clarified if they include the articular cartilage part or not.
- L.38-40, "We further show that reparative Gli1+ cells originate from Pdgfra+ cells outside the cartilage, revealing the surrounding tissues as an unexpected CP source". --- Note that contribution of the surrounding tissues, such as perichondrium and Ranvier's groove, to cartilage formation has been previously indicated, thereby it may not be 'unexpected'.
- Figure 5b", most tdT+m cells are positive to EdU+ while b' shows only 10-25% cells are double stained. Explain this discrepancy.
- Figure 7f. Considering the cartilage marked by the authors (ex, Extended Data Figure 6d), the articular cartilage region should be possible source in addition to the groove of Ranvier.

Reviewer #2

(Remarks to the Author)

This revised version fully addresses the comments from the initial version.

Reviewer #4

(Remarks to the Author)

Version 2:

Reviewer comments:

Reviewer #1

(Remarks to the Author)

Most of the concerns have been addressed, but two questions remain:

1. Add explanation how to set the threshold in the ApplyThreshold module in CellProfiler. If this is not the Automatic method, explain the manual methods how to input a specific threshold value.
2. Some data sets have $n=2$ (Figure 5d, 5d"; Extended Data Figure 6f, several groups, Extended Data Figure 10b', CTR). It is unlikely that these samples would pass a Shapiro-Wilk normality test. Check this. If they do not, the authors need to conduct a nonparametric analysis, or omit it from the statistical analysis to ensure scientific rigor. The differences are clear, it would not change the conclusions of the manuscript.

Rebuttal letter for NCOMMS-24-81312A

We thank the reviewers for their reasonable and useful feedback. We think that addressing their comments has significantly improved the manuscript and reinforced its conclusions. Please find below a point-by-point answer.

Reviewer #1

1. Clarification of transgenic mouse strains. Clarify the mouse strains of all transgenic mice used in this study, and state how the strategy of transgenic mouse generation minimizes such experiment flaws. Gli1CreER mice have been made by insertion of CreER into the Gli1 locus at the 1st exon, and Southern blot analysis has shown a shorter size of Gli1 recombined allele (Cell. 118, 505-516, 2004). Confirm that the CreER insertion does not disturb endogenous Gli1 expression and the use of homozygous Gli1CreER mice for breeding is appropriate.

The combination of multiple transgenes is a necessity of this type of experiment. As the reviewer points out, it is important to use adequate controls in the multi-transgene experiments. This is why the crosses have been designed so that every litter has control and experimental mice, with controls being very similar to experimental ones, except for one transgene (for example, having *Gli1-CreER*, *TRE-p21* and *tdTom* reporter, but not *Col2a1-rtTA*). For this Refined experimental design to produce a reasonable number of useful mice, and thus being able to Reduce the number of mice being used (achieving two of the 3Rs principles), we need to use parents that are homozygous for all alleles except for the one that distinguishes control and experimental mice. And this is why we use *Gli1-CreER* homozygous breeders, so that 100% of the offspring is *Gli1-CreER* heterozygous. Of note, this is a null allele that has been previously characterised. The senior author did his postdoctoral research in the Joyner lab, the group that generated the *Gli1^{CreER}* mouse (which is a null allele), and there it was shown that homozygous animals are viable and fertile, although males have somewhat reduced fertility (Park et al. 2000, PMID 10725236, see also JAX entry for this line). Since we used females to generate our experimental litters, we combined the advantages of having normal fertility and 100% of useful offspring.

2. The study uses two types of cell-cycle arrest models, Left-Cart-p21 and Pan-Cart-p21. The phenotypes of these two models are different regarding the left-right discrepancy and completion of catch-up. The Left-Cart-p21 shows growth retardation (Ext.Fig.2c) while the Pan-Cart-p21 growth plate seems to catch up completely (Fig.1b'). The compensatory mechanism for cell-cycle arrest in these two models may be similar to a greater or lesser extent, but may differ from each other. Combining the results of the two models is therefore complex and difficult to interpret accurately. As the current study starts from previous results obtained from Left-Cart-p21, the authors could have included the results shown in Figure 1a, but in the remainder of the manuscript it would be better to focus exclusively on the Pan-Cart-p21 model.

One clarification: the *Left-Cart-p21* model does not show growth defect in the tibia, and only a biologically minimal (even if statistically significant) defect in the femur. However, we agree that the models may differ in some of the compensatory mechanisms and that having both models makes the paper more complex. In the interest of clarity, we have removed the unilateral model after Figure 1, as suggested.

3. To clarify whether the increase in the number of Gli1⁺ and Pdgfra⁺ cells is due to de novo Gli1/Pdgfra expression, an expansion of lineage cells or both, and where these de novo expressing cells reside, it is necessary to examine tempo-spatial changes in gene expression of Gli1/Pdgfra in the lineage-tracing models. [...] Gli1 expression in Gli1-lineage tracing model and Pdgfra expression in Pdgfra model should be included. It would be also informative to show the relation among p21 expressing cells, Ki67-cells, Ccn2 expressing cells and Gli1-expressing and lineage cells to understand which cells respond to cell-cycle arrest.

Thank you for the suggestions. We have characterised *Gli1* expression in the Gli1-lineage model and *Pdgfra* expression in the Pdgfra-lineage model, as requested. This showed that, after short tracing, neither *Pdgfra*⁺ cells, nor Pdgfra-lineage cells, were found in the cartilage (Ext. Data Fig. 5c). In addition, in normal growth, *Gli1* is downregulated in a significant proportion of the E13.5-traced chondrocytes, whereas upon p21-misexpression, the proportion of Gli1⁺ cells within the traced lineage increases significantly (Ext. Data Fig. 5a'). Moreover, our analysis showed that when the Gli1 lineage was marked 1d before p21 induction, there was a significant increase in the number of Gli1⁺ cells that did not belong to the traced lineage (Ext. Data Fig. 4g). This result supports the idea that while some Gli1⁺ cells may

retain *Gli1* expression in response to Cart-p21, $Gli1^-$ cells can also become $Gli1^+$ upon challenge.

Regarding which cells respond to cell-cycle arrest, we already showed that it is the $p21^-$ cells that show enhanced proliferation in p21-expressing cartilage (Rosello-Diez et al. 2018, PLoS Biol). To complete this type of analysis, we have now calculated the proportion of $Ccn2^+$ cells in the $Gli1$ -lineage and in the non- $Gli1$ lineage, and in both populations the decrease in $Ccn2$ expression in EXP samples is comparable (Fig. 7c, c'), indicating that $Ccn2$ downregulation is not due to a cell-autonomous effect of $Gli1$ upregulation. Moreover, we used the Distance Analysis tool DiAna and found that $Ccn2^+$ cells in experimental samples tend to be further away from $p21^+$ chondrocytes than from $p21^-$ ones (Ext. Data Fig. 10c). This suggests that $p21^+$ chondrocytes generate a “ $Ccn2$ -inhibitory area”, an intriguing result that we will explore in future works. Lastly, analysis of $Ki67^+$ and $Gli1$ -lineage cells (Ext. Data Fig. 4c) showed that the $Ki67^+$ pool (cycling chondrocytes) gets enriched in $Gli1$ -lineage cells in response to p21 expression in the cartilage.

4. Fig. 1c, c' and Ext Fig 4. There is no evidence that tamoxifen clearance is within one day. It is likely that tamoxifen remains active in the body for more than one day. At least Tamoxifen should be injected in prior to Dox treatment. The data do not support the statement “This result suggests that while the existing population of $Gli1^+$ cells contributes to the repair, some $Gli1^-$ cells become $Gli1^+$ after p21 induction in the cartilage” (lane143-144). As discussed above, it is more convincing to address upregulation of de novo $Gli1$ expression by demonstrating upregulation of $Gli1$ transcripts in non- $Gli1^+$ cells.

This is a very important point, which has been previously addressed. A study by the Mackem lab (Nakamura et al. 2006, Dev. Dynamics) analysed the dynamics of CreER activation in the cartilage upon tamoxifen injection, finding the following: “tamoxifen was injected at different time intervals ranging from 8 to 36 hr before harvesting embryos [...] LacZ activation was first detected in scarce cells at 8 hr and considerable mosaic recombination was already evident within 12 hr of injection. Between 16 and 24 hr, extensive recombination was observed, approaching completion by 24 hr [...] Panel2E: Recombination, as judged by the percent LacZ+ cells, was essentially complete by 24 hr”.

Moreover, we have done the requested experiment (Tam at E12.5, Dox at E13.5, new Ext. Data Fig. 4d, e, g, h), and the results supported our previous conclusion. Namely, expansion of the $Gli1$ lineage was less pronounced in this case (as compared to Dox E12.5, Tam E13.5). Moreover, in response to a previous question we have shown expression of *Gli1* in cells that were not labelled after Tam injection at E12.5, further supporting the de novo activation of *Gli1* expression in response to cell-cycle arrest in the cartilage.

Together, the results indicate that part of the response to p21 is due to $Gli1$ -negative cells, which can only be followed if Tam is injected 1d after Dox.

5. Fig. 2. $Gli1$ -Cart-DTA group is needed as additional control. As shown in Fig 5b', growth plate function in the Gli -Cart-DTA mice was seriously affected. DTA ablation would affect not only compensated $Gli1^+$ cells but also other $Gli1^+$ progenitors present in the growth plate, and Fig 2 experiments may not demonstrate a specific role for $Gli1^+$ cells in compensating cell-cycle arrest. “This result demonstrates that $Gli1$ -derived chondrocytes are required to compensate for cell-cycle arrest in the fetal cartilage” (lane 162-164) seems overestimated.

Of note, the *Gli-Cart-DTA* group was generated in the initial crosses that we set up for the experiment, but not included in the original manuscript for simplicity. We have included it now (New Fig. 2b' and c). While the *Gli-Cart-DTA* group shows a trend towards diminished bone length, at P0 this trend is not significant as compared with controls or *Pan-Cart-p21* mice (over time it does become significant, as we now show in Fig. 4). Since neither *Pan-Cart-p21* nor *Gli-Cart-DTA* show a significant effect on bone length individually, but they do when combined, our statement that “ $Gli1$ -derived chondrocytes are required to compensate for cell-cycle arrest in the fetal cartilage” is not exaggerated. We have however added the following sentence to acknowledge the limitations of the approach: “... $Gli1$ -derived chondrocytes are required to compensate for cell-cycle arrest in the fetal cartilage, although it did not rule out more subtle effects in *Gli-Cart-DTA* pups, which showed a trend towards decreased bone length. Later time points could not be analysed due to the high mortality of animals carrying the *Gli1-Cart-DTA* genotype”.

6. Fig. 4. The authors show that most GL retaining cells were $Gli1$ -lineage, indicating that Gli -lineage cells

contain LLCPs. They also show that the GL retaining cells decreased in the Gli1-lineage in the RZ, and conclude that GL-unlabeled Gli1-lineage progenitors are newly added Gli1-lineage cells from outside the growth plate. However, to reach this conclusion, there are at least two different interpretations that must be ruled out.

(1) Extended Fig. 6C shows that the cells at the top of the cartilage did not express GL, but Gli-lineage cells (Tom⁺), suggesting that even in the growth plate, some Tom⁺ cells are not GL-labeled, especially in the RZ. If these non-labeled LLCPs produce progenitors in the RZ, the GL-retaining population in the Gli1-lineage should decrease.

To test this possibility, we performed co-expression analysis on the specimens mentioned by the reviewer. Besides GL and tdTomato, we used hybridisation chain reaction to assess expression of *Matn1*—a cartilage marker independent of *Col2a1*, as *Col2a1* regulatory region is used to drive rtTA and hence nGL (Ext Data Fig. 6d). We could not detect almost any tdTom⁺ nGL⁻ cells within the *Matn1*⁺ region (except at the connection between tibia and fibula, a region that is not expected to contribute to bone growth), strongly suggesting that the dilution of GL⁺ cells within the Gli1 lineage after a 1d Dox-pulse is not due to the mechanism hypothesised by the reviewer. Of note, this is not the only experiment in the manuscript that supports the ingression of new waves of chondroprogenitors from outside the cartilage (see *Pdgfra*-lineage results).

(2) Once the Gli1-lineage LLCPs were labelled by GL, and undergo a certain number of asymmetric divisions to produce progeny, their GFP intensity may decline beyond detection at P21 or later, and only quiescent GL-LLCPs may retain GL until P21. Therefore, the reduced GL-retention in Gli1⁺ lineage cells could be due to dilution of the label by cell division, rather than "newly added" Gli1 lineage LLCPs.

Yes, this is exactly what we meant. The decrease in GL⁺ cell number after P14 is indeed most likely due to cell division of the previously quiescent cells. The point we were trying to make is that a higher number of label-retaining cells is detected in the cartilage when comparing Dox E12.5-E18.5 with DoxE17.5-E18.5. Since the end-point of Dox exposure is the same in both cases, the most parsimonious explanation is that there is continuous addition of LLCPs from outside the cartilage between E12.5 and E18.5.

7. *Pdgfra*-related experiments have serious concerns regarding the statement "we chose platelet derived growth factor alpha (*Pdgfra*) as a marker of stromal cells that is not expressed in the cartilage" (lane 245-246). This reviewer is concerned that, judging from the data given, the *Pdgfra* lineage cells do not necessarily come from outside the growth plate, and that *Pdgfra*-related experiments may not be of significant relevance in this manuscript.

While we respectfully disagree with the last statement, we understand the reviewer's reservations, and have provided additional evidence as suggested (see below). We think that, now, these experiments strongly support new biology that will open new avenues of fundamental and clinical research.

Some chondrocytes are clearly positive to *Pdgfra* in Ext Fig. 5b. Ext Fig. 5b' image (likely *Pdgfra*-GFP reporter) shows that some cells look positive when the image brightness is increased a little bit. To support the statement "the surrounding tissues can contribute to cartilage" (lane 250-251), it is necessary to provide the results that no *Pdgfra*⁺ cell is present in cartilage in the *Pdgfra*;R26LSL-tdT mice at E13.5 after Dox injection at E12.5. Higher resolution (higher quality) of in situ hybridization image of *Pdgfra* from E12.5 to E14.5 should be provided to prove that *Pdgfra* expression is exclusive in tissues outside the cartilage.

We have addressed this comment in two ways, as suggested: i) short-term tracing of *Pdgfra*-lineage from E12.5 to E13.5, combined with analysis of *Pdgfra* RNA expression; ii) high-resolution in situ hybridisation of *Pdgfra* on hindlimb sections at E12.5, E13.5 and E14.5. Experiment i) showed that this short-term tracing led to barely any tdTom⁺ or *Pdgfra*⁺ cells inside the cartilage (Ext. Data Fig. 5c). Experiment ii) showed again that *Pdgfra* mRNA is absent from the E13.5 and E14.5 cartilage elements (identified based on cell morphology and density), and that only at E12.5 was there a faint gradient of expression in the periphery of the cartilage element (Ext. Data Fig. 7). Together, these experiments suggest that *Pdgfra* is almost only expressed outside the cartilage, and is downregulated as cells get incorporated into the cartilage around E12.5. Moreover, we have also inverted the Dox and TM regimen, to create two different scenarios. When the *Pdgfra*-lineage was labelled first and p21 expression activated later, there was a significant increase in the proportion of *Gli1*⁺ cells within the *Pdgfra*-lineage (Fig. 6c triangles). However, when the *Pdgfra*-lineage was labelled after p21 induction, besides the increase in *Gli1*⁺ cells within the *Pdgfra*-lineage (Fig. 6c' triangles), there was also an increase in the number of *Gli1*⁺ cells that did not belong to the *Pdgfra*-lineage (Fig. 6c' squares). This supports the idea that *Pdgfra*⁺ cells in the

perichondrium progressively transdifferentiate into Gli1⁺ chondroprogenitors once p21 expression is activated in the cartilage (see graphical model in Fig. 6d).

Fig. 6a shows Pdgfra-TdTom positive cells are more than 30% in the control cartilage at E14.5. In Fig.6d', in the control group, PDGFRA⁺ cells were observed in the growth plate at E14.5. The data are open to several interpretations: 30% of chondrocytes are added from the Pdgfra⁺ cells in tissues outside the cartilage, Pdgfra expression is induced inside the cartilage, Pdgfra⁺ cells are present in cartilage already, or a mixture of these events occurs.

It is important to note that, in this case, what we detected in Fig. 6a (now Fig. 6e, e') was PDFRA protein, not RNA. In fact, our trajectory analysis based on snRNA-seq (Ext. Data Fig. 5d), our colorimetric in situ hybridisation (Ext. Data Fig. 7) and our HCR data (Ext. Data Fig. 5c) all show that there is barely any expression of *Pdgfra* mRNA in the cartilage. Thus, we hypothesised that *Pdgfra* is downregulated as the cells ingress in the cartilage, although PDGFRA protein remains for a while. We indicated this in the original manuscript: "we reasoned that chondrocytes that had recently formed from Pdgfra⁺ cells in response to cell-cycle arrest would retain PDGFRA protein for some time". We have modified this sentence to add clarity: "Although Pdgfra mRNA levels are very low in the cartilage (Ext. Data Fig. 5c-d, Ext. Data Fig. 7), we reasoned that chondrocytes that had recently formed from Pdgfra⁺ cells in response to cell-cycle arrest would retain PDGFRA protein for some time".

Minor specific comments

1. "fetal-cartilage targeted cell-cycle arrest" in the abstract is difficult to understand without reading the main text.

Thank you, this has been changed to "genetically-induced cell-cycle arrest targeted to the fetal-cartilage".

2. Fig 1a'. "(GLI1) transcription factor was significantly upregulated in the left experimental (ExpL) osteochondroprogenitors and to a lesser extent in resting chondrocytes". In Fig.1a', it seems that Gli1 expression is more prominent in resting chondrocytes. To demonstrate the difference between control and experimental groups, other visualization method such as fold change, or a split violin plot would be better.

In that sentence we referred to changes in regulon activity between Exp and Ctl samples, rather than absolute levels. In any case, we have now used violin plots to show GLI regulon activity, to allow for better comparison (New Fig. 1a' and Ext. Data Table 2). A partial DotPlot is shown in Ext. Data Fig. 3a, and the data underlying the full DotPlot is located in Ext. Data Table 1.

3. Fig 1c'. The statement "in control conditions, most chondrocytes derived from fetal Gli1⁺ cells are only transient and eventually eliminated from the cartilage"(lane 127,128) is somehow incorrect because Fig.1c' shows a robust population of Gli lineage cells stayed in the cartilage at P0 compared to E14.5

Yes, the sentence is not accurate. We have changed it to "in control conditions, some chondrocytes derived from fetal Gli1⁺ cells are only transient and eventually eliminated from the cartilage".

4. Fig 1c, c'. The trabecular area should not be included in the "outside" population because there should be a substantial lineage⁺ population which are derived from the Gli1 lineage chondrocytes (i.e. transdifferentiation).

Thank you for the suggestion. We have redefined the "outside" region and modified Fig. 1c, c' accordingly. The conclusions remain very similar.

5. Fig.3. "short-lived clones that get depleted over time, and long-lived ones that remain dormant at perinatal stages and become more active after the second postnatal week" (lane188-190). It is possible that the same Gli1 lineage progenitors give rise to short columns at earlier time points but start to produce long columns at the second week. There is no convincing evidence showing that the short-lived progenitors were depleted. There are a certain number of GLI1⁺ cells in RZ without making column. How do the authors interpret them? This population seems quiescent, and would be GL retaining cells in the Gli1-lineages cells in the RZ as described in the major comment 6 above.

Thank you for the observation. Cell behaviours are rarely deterministic, as there is always a degree of

stochasticity. In other words, if the probability of division of a long-lived CP in the early cartilage is very low, then over the course of the last prenatal and first postnatal week it will divide once at most. Thus, our interpretation of the reviewer's observation is that, at perinatal stages, long-lived CP divide very few times (likely only once) asymmetrically, giving rise to i) a short-lived progenitor that will continue dividing for a while, producing a short transient column, and ii) a quiescent progenitor that will remain inactive until late postnatal stages. To further test this possibility, we provided saturating levels of EdU at E13.5-14.5 and then quantified the number and spatial location of EdU-retaining cells up to P3, P7 or P14 (Ext. Data Fig. 5b-b"). We found that the Gli1-lineage in the RZ gets enriched in EdU-retaining cells after P7, indicating that the non-EdU-retaining ones (i.e., the ones that form short columns) are eliminated from the RZ.

6. Chondrocytes generated in the last postnatal week from Gli1+ cells are required for long-term bone growth (lane 231-232). The paragraph is not consistent with this section title.

Apologies, we meant prenatal. This has been corrected.

7. Fig.6b', c. These figures clearly show that the percentage of Pdgfra-lineage cells in Gli1-expressing chondrocytes decreased upon p21 induction, and there was a robust expansion of Gli1+ cells within the growth plate. These results suggest that some Gli+ cells are not derived from the Pdgfra lineage. To validate the significant contribution of Pdgfra lineage to the compensatory mechanism, the authors might perform DTA experiment in these cells.

While this experiment is not minor, we happily undertook it given its relevance. As shown in Ext. Data Fig. 8c-c', ablation of Pdgfra-derived chondrocytes in the presence of p21 impeded most of the increase of Gli1+ cells. There were residual Gli1+ cells, indicating indeed that not all Gli1+ cells are Pdgfra-derived.

8. Fig 7e. If the authors would like to argue the involvement of Ccn2 in addition of Gli1/Pdgfra+ cells into the growth plate, they need to provide evidence that down-regulation of Ccn2 stimulates or up-regulation of Ccn2 inhibits cell migration from the perichondrium (groove of Ranvier) in ex vivo or in vivo.

We don't make a claim about the role of Ccn2 in migration. We have some candidates in this regard, but their validation would be beyond the scope of this manuscript.

9. Discussion. Previous reports (Nature 2013 499:491-5 and J Bone Miner Res 2019, 34:964-974) have performed the lineage tracing experiments with CtskCre mice and shown that a only a small proportion of cells in the groove of Ranvier constitute the growth plate, but contribute little to making columns of the growth plate. The authors should discuss these findings.

Thank you for the suggestion. We have included those references in following part of the Discussion: "It is worth noting that previous studies used *Ctsk-Cre* to label most cells of the perichondrial groove of Ranvier, finding little³⁶ to no³⁷ contribution to the growth plate from P14 or P7 onwards, respectively. This aligns well with our finding that, in the absence of challenge, the Pdgfra-lineage almost did not contribute to the central growth plate from P14 onwards (Ext. Data Fig. 8c)".

Reviewer #2 (Remarks to the Author)

In this study, Qu and colleagues ask how embryonic limbs are able to compensate their growth when injured by a loss of proliferative cells. Using sophisticated genetic mouse models that enable mosaic cell cycle arrest in the left limb only, they show that descendants of Gli1-expressing cells are necessary for this compensation, and in the absence of compensation serve as an important progenitor cell population. Fetal Gli1+ cells are a population of label retaining progenitor cells that give rise to chondrocytes during postnatal bone development and are necessary for normal bone length. Upon injury, *Pdgfra*-expressing cells give rise to Gli1+ expressing cells, identifying a mechanism for amplification of this population to compensate for growth in the injury model. Finally, they find that CCN2, a negative regulator of bone growth, is inhibited in *Cart-p21* MOE experimental samples, suggesting a possible mechanism for the expansion of Gli1+ cells.

Overall, this is an exciting and novel study. The experimental design has multiple strengths, including sophisticated, novel genetic models, rigorous experimental controls and well-designed experiments with appropriate replicates and quantification. The findings of a fetal progenitor cell population are novel and will serve as a foundational study for understanding how this progenitor population is regulated both during normal bone development as well as in growth compensation. Although I do not have any major concerns, several minor concerns are listed below for the author's consideration.

1. Although the Shi et al, 2017 study implicating Gli1+ cells in fracture repair is cited, additional work from Francesca Mariani's lab has also shown that Hedgehog signaling is essential for fracture repair, additional discussion of this is warranted in the discussion. For example, the Mariani lab has found that Hedgehog signaling is important in a rib fracture model Kuwahara et al, 2019 *Elife*8, e40715. Please consider a brief addition to the discussion that considers these findings with the newly defined Gli1+ fetal progenitor cells in this study.

Thank you for the suggestion. The reason we did not include this reference in the original manuscript is because hedgehog does not seem involved in the observed upregulation of *Gli1* in our model. If anything, *Ihh* is downregulated along with other markers of prehypertrophic chondrocytes. Nevertheless, we agree that it is worth discussing the similarities and differences between these studies, so we have integrated a discussion of Kuwahara 2019 into the paragraph that starts with "Lastly, regarding the signalling changes operating in the challenged limbs..."

2. Consider replacing Fig 1a (which on its own is confusing) with Extended data Fig 2a, which greatly clarifies the experimental foundation.

We agree and have done this. Thank you for the suggestion.

3. Figure 1a'. The enrichment of Gli1 activity by SCENIC is confusing in its presentation. Are the depicted regulons shown in order of enrichment – and by what criteria? If not, how are they picked? Is this tested using experimental only (left versus right) is this a multiparametric analysis? Are all pairs tested for significance with only Gli1 reaching significance (in the outlined and dashed domains) or are other domains significantly enriched as well? Please also clarify the abbreviations used in the figure panel so that they are consistent between what is described in the text and on the figure image (e.g. 'ExpL' vs 'EL').

We have moved this DotPlot to Extended Data Fig. 3, and replaced it with a Violin Plot in Fig. 1a', focussed on the Gli1 regulons only. The DotPlot is not presented in any ranked manner but, as we now explain in the text, was filtered based on our previous bulk RNA-seq analysis of the *Left-Cart-p21* model (Rosello-Diez et al. 2018), which led to Gli1 as the candidate regulon most supported by the combined data. We

have also changed all appearances of EL and ER to ExpL and ExpR, and made sure that this was consistent across text and figures.

In addition to Gli1 regulon activity, were Gli1 or Gli1-regulated network genes differentially expressed between the left and right knees of the Left-Cart-p21MOE mice? Even though the subsequent data makes a compelling case that Gli1+ cells provide compensation in growth-challenged limbs, this first evidence seems tenuous as presented.

To better support the choice of Gli1, we have provided Violin Plots using the Regulon score, and included statistical tests (Fig. 1a'). Of note, we also examined the expression patterns of genes within the network. Gli1 showed the most prominent changes, though other genes—including Negr1, Ptch1, Ntn1, and Nup37—also displayed differential expression across conditions, consistent with bulk RNA-seq data. Given that Gli1 and Ptch1 belong to the same pathway, we focused our analysis on Gli1 as the key transcriptional effector.

4. Figure 7A. Does this show the most significant DEGs (as measured by FC/FDR?) or is this a curated subset? If the latter, please provide a statement about where Ccn2 fits into this overall set of DEGs.

We have replaced this panel with a ranked heatmap (using z-scores), to provide clarity on the position of Ccn2 within the DEGs. This is now clarified in the text: “Besides the expected cell-cycle related genes (e.g. histone-encoding), the second-biggest change was observed in a gene related to chondrocyte activity and response to stress, namely cellular communication network factor 2 (Ccn2)”.

Below, we also include other relevant metrics (e.g., fold change, FDR) to provide additional context.

gene	logFC	AveExpr	t	P.Value	adj.P.Val	B
Hist1h2ae	-2.13252	8.158306	12.31777	1.12E-07	0.001531	8.114702
Matn3	-1.95929	7.073822	10.77517	4.27E-07	0.002921	6.851227
Hist1h4d	-2.08959	5.520353	9.544465	1.40E-06	0.0048	5.594296
Hist1h1b	-2.17035	6.404491	9.011092	2.44E-06	0.006688	5.220236
Gm42743	-2.93141	3.616026	9.712004	1.19E-06	0.0048	4.663877
Hist1h3c	-2.64756	4.56901	8.645766	3.63E-06	0.008274	4.51442
Ccn2	-2.23673	6.104922	7.672669	1.11E-05	0.021698	3.798239
1700001K19Rik	-2.3458	4.16753	7.458849	1.44E-05	0.024611	3.25682
Hsph1	-1.98731	6.232323	6.935725	2.78E-05	0.040011	2.933964
Hist1h2an	-2.81915	3.735908	6.742845	3.57E-05	0.044393	2.327934
Hist1h2bb	-2.60431	2.89787	6.895733	2.92E-05	0.040011	2.00446
Ctnnd2	1.2601	4.928998	-5.91011	0.000111	0.044868	1.595086

5. Figure 7: Although there is interesting data presented on Ccn2, Ctnnd2 (delta catenin) appears highly upregulated in the experimental Cart-p21MOE samples compared to controls. Please consider commenting on this and how it might fit into models for Gli-lineage activation.

Indeed, we also were quite intrigued by this candidate. Unfortunately, all our attempts to confirm changes in *Ctnnd2* expression failed, as we were unable to detect any signal using either colorimetric in situ hybridisation or HCR.

6. Figure 7C: I understand this figure from the description in the text but I'm confused about the label 'Asmu-Swiss'. Isn't this the Pan-Cart-p21MOE model?

It is indeed the *Pan-Cart-p21^{MOE}* model. These embryos were generated by crossing *Col2a1-rtTA; TRE-p21/p21* mice with WT (Swiss Webster) mice, hence the confusing label in the figure panel. We have amended this panel to only show potential embryo genotypes, instead of the cross.

7. Methods section lines 685-700 what seems to be a redundant, less detailed description of the HCR protocol that should be deleted.

Done, thank you.

8. Extended data. With the exception of Extended Data Fig. 5, it was a bit challenging to figure out which figure was which since they aren't numbered and don't have embedded legends. Consider adding Figure numbers and, if allowable by the journal, embedded figure legends.

We fully agree with adding Figure numbers, which we have done. We believe that Figure Legends are only to appear on the text.

Reviewer #1 (Remarks to the Author)

The revised manuscript has been improved by addressing the concerns. However, several important points remain to be clarified. There are also additional suggestions.

1. 1. Transgenic mouse strains. Please provide information about the background strains of transgenic mice used by the authors for breeding. Mixing background strains will result in variability in growth and characteristics. Background information is important for scientific rigor and will be useful for future experiments.

Thank you for the suggestion. When this study was started, most mouse lines were obtained outbred to a Swiss Webster background, so the same was done for new lines such as the RGBow, for consistency. Outbred backgrounds show more robust growth properties, and this situation was considered preferred to the uncertain outcome of mixing different backgrounds. A sentence about genetic background has been included in the Methods section.

2. The origin of Gli1 and Pdgfra-lineage cells.

Extended Data Figure 5c. This reviewer agreed that a majority of Pdgfra transcripts and Pdgfra-lineage cells were outside the cartilage where the authors marked by the white-dotted line. However, Pdgfra transcripts (white) and Pdgfra-lineage cells (weak red signals) were also found inside the white line and likely double-labeled with GL (green). The authors should not ignore these cells. Therefore, the possibility remains that some Pdgfra-expressing and -lineages cells may exist inside the cartilage. Note that this reviewer does not mean to deny the author's idea that Gli1- and Pdgfra-lineage cells are added outside the cartilage and that challenging increased number of Gli1-/Pdgfra-lineage cells. Indeed, addition of chondrocytes from the surrounding tissue such as Ranvier's groove has been indicated previously.

We agree with the reviewer's observation of some *Pdgfra* transcripts present at E13.5 but, as shown in Ext. Data Fig. 7, *Pdgfra* expression is quickly downregulated in the cartilage at/after E13.5. We have modified lines 260-261 and 288-289 to make sure this is clear:

“(Pdgfra) as a marker of stromal cells that is virtually not expressed in the cartilage beyond E13.5 (Ext. Data Fig. 5c-d, Ext. Data Fig. 7)”

“Although Pdgfra mRNA levels are low and transient in the cartilage (Ext. Data Fig. 5c-d, Ext. Data Fig. 7), we reasoned that chondrocytes that had recently formed from Pdgfra+ cells in response to cell-cycle arrest would retain PDGFRA protein for some time”

Extended Data Figure 6d and L.266-268 “As shown in Ext. Data Fig. 6d, we could not detect almost any tdTom+ nGL- cells within the Matn1+ region, ruling out that possibility.” --- This statement is not agreeable at this moment. Please provide gray scale single raw images. It is difficult to monitor exact positive signals of composed images. It seems that Matn1 transcripts (Magenta) and nGL-cells exist outside/above the white dotted line. The nGL system and Matn1 expression may label broader areas than the authors define as cartilage.

We have provided the separated grayscale images. There are indeed Matn1+ nGL- cells in extremes of the epiphysis (regions that do not contribute to growth, see new Ext. Data Fig. 6d) so, if anything, this would go against the reviewer's interpretation that nGL labels a broader area than the cartilage.

3. Tamoxifen and Dox efficiency and clearance

L.146-148, “, with TM clearance mostly taking place before any potential de novo Gli1 expression. Indeed, it was previously shown with a cartilage-driven CreER that recombination is complete 24h after TM injection¹⁹. - Reference 19 shows that TM is sufficient to induce Cre recombination within 24 hours by monitoring LacZ expression, but does not show how long it takes for TM to be cleared.

We respectfully disagree. Reference 19 does say that the % of labelled cells did not increase after 24h, implying that TM is cleared after that.

Figure 4c. The extension of the Dox administration period increased the number of tdTom-negative/nGL-positive cells in the RZ, suggesting that the extension of the Dox administration period may increase the number of labeled cells even inside and the cartilage. --- The experiment in Figure 4 is based on the idea that 24 hours of drug administration fully activates cre recombination and drug clearance is achieved within 24 hours. Although high efficiency of tamoxifen administration is demonstrated, the mouse system used in this study is not exactly the same as the study cited by the authors, and Dox efficiency and clearance are not evident. If one day of Dox administration does not achieve full efficiency and several days of exposure are required, a simple interpretation of the data as the authors did would be not made. The authors need to consider more carefully the uncertainties in clearance and efficiency of TM and DOX in in vivo experiments.

It is worth noting that we previously reported that Dox treatment from E12.5 to E16.5 leads to activation of TRE-driven transgenes targeted to the *Tigre* locus in 80% of cartilage cells (Ahmadzadeh et al. 2020). Here, with Dox given only from E12.5 to E13.5, already 70% of cartilage cells show nGL expression (Ext Data Fig. 6e), indicating that activation is almost complete. Moreover, the important result is that the same Dox-pulse duration leads to a quantitatively different outcome when the pulse is given at E17.5 as compared to E12.5, suggesting that the labelled populations do not fully overlap. In any case, we have added one sentence to acknowledge the uncertainty of full Dox-dependent activation after 1 day: “In other words, while one-day Dox pulse may not be enough to fully activate nGL expression in all possible rTA+”

cells, the same pulse duration, given at E17.5, led to a different quantitative outcome at P21 as compared to E12.5".

4. Image analysis:

- - The authors' determination of negative/positive signals in images, such as the Extended Data in Figure 6C, raises the question of how to define the signal threshold. Please provide a more detailed method beyond the use of CellProfiler, since cell analysis is one of the most important parameters in this study.

Thank you for the suggestion. Manual curation was used for every single image, by checking whether the outlines of the automatically identified objects matched the eye of the curator, who was blinded to the genotype. We have added these details to the Methods sections, and uploaded an example of a CellProfiler pipeline, with explanatory notes, as supplementary material.

- The authors state that "Non-parametric tests were selected when the assumption of normality could not be met." It is likely that all data (comparison of bone length and cell analysis) are statistically analysed by parametric one way or two way ANOVA. Provide the method how to determine normality.

Apologies, that sentence was left inadvertently. Indeed, only parametric tests were used (Shapiro-Wilk test for normality was used when in doubt).

Other modest to minor comments

- The definition of the resting and proliferating cells in E14.5 and E17.5 tissues is unsure. Looking at the histology shown in Extended Data Figure 7, especially in E14.5 limbs, the difference between resting and proliferating chondrocytes is difficult to be defined by "The transition between round (resting) and flat (columnar) nuclei, forming an arch along the upper point of the grooves of Ranvier, was considered as the start of the PZ". The top border of the RZ is also unsure. The top border of RZ needs to be clarified if they include the articular cartilage part or not.

We have clarified that the RZ is considered to include the articular region (except the outer-most 2 cell layers) at the stages before a secondary ossification centre exists. Finding the RZ-PZ transition may become challenging at early stages, but using the objective criteria that we describe allowed for consistency across observers.

- L.38-40, "We further show that reparative Gli1+ cells originate from Pdgfra+ cells outside the cartilage, revealing the surrounding tissues as an unexpected CP source". --- Note that contribution of the surrounding tissues, such as perichondrium and Ranvier's groove, to cartilage formation has been previously indicated, thereby it may not be 'unexpected'.

We respectfully disagree. At least in the case of mouse, the contribution of the perichondrium to the cartilage has never been shown as clearly as we have. In line 390, 391, we discuss two papers that used different tools to lineage trace some populations of the groove of Ranvier, and they did not find long-lasting contributions to the growth plate.

- Figure 5b", most tdT+m cells are positive to EdU+ while b' shows only 10-25% cells are double stained. Explain this discrepancy.

We guess the reviewer means Ext. Data Fig. 5b" and b'. b' is EdU+ cells within the tdT population, while b" is CldU+ cells within the EdU+ population. These are not comparable, of course.

- Figure 7f. Considering the cartilage marked by the authors (ex, Extended Data Figure 6d), the articular cartilage region should be possible source in addition to the groove of Ranvier.

Agreed, we have added this to the model and the legend.

Reviewer #2 (Remarks to the Author):

This revised version fully addresses the comments from the initial version.

Excellent, thank you.

Reviewer #4 (Remarks to the Author)

Great initiative, thank you.